# Causal-aware Anomaly Detection for Tabular Data

**Dang Nguyen** [* 1]   **Tu Anh Hoang Nguyen** [* 1]   **Thuc Duy Le** [2]   **Svetha Venkatesh** [1]   **Trung Le** [3]   **Sunil Gupta** [1]

## Abstract

Anomaly detection (AD) methods often ignore causal dependencies and treat anomalies as outliers, which is brittle when anomalies are primarily mechanism violations rather than extreme values. We propose CausalAno, a causal-aware detector that trains a causal GAN on normal data and leverages its discriminator to learn mechanism-consistent representations. We score test samples by fitting a Gaussian model in this feature space and computing the Mahalanobis distance, measuring deviation from the normal causal manifold. We demonstrate the effectiveness of CausalAno with extensive experiments on 28 tabular datasets (18 continuous-only and 10 mixed-type), comparing against 16 SOTA baselines. Our results show consistent improvements across both mixed-type and numerical-only settings. Our ablation studies further confirm that the gains come from the causal factorization in the causal GAN rather than a generic GAN backbone. Our CausalAno offers a practical and effective solution for real-world applications where anomalies often arise from mechanism-violating behaviors.

## 1. Introduction

Anomalies present a persistent challenge in real-world tabular data, where even a small fraction of abnormal records can distort predictive models and disrupt critical operations (Huang et al., 2025; Kumari et al., 2024). Effective detection of such anomalies–in fraud screening, intrusion detection, and clinical alerting–enables earlier risk mitigation, reduces financial losses, and strengthens regulatory compliance. For example, in mental health monitoring, anomaly detection (AD) applied to tabular behavioral profiles–such as sleep duration, heart rate, and mobility–can identify atypical deviations that serve as early warning signs for depressive

relapses (Zhang et al., 2025). Because anomaly labels are typically scarce, noisy, or expensive to obtain, most practical AD methods operate in *unsupervised* setting i.e., only normal samples are available during training.

Among existing approaches, a widely used and competitive family is reconstruction-based methods (Shyu et al., 2003; Li et al., 2022; Pang et al., 2018; Liu et al., 2021). These methods learn normal patterns from unlabeled data by encoding samples into a latent representation and decoding them back to the input space so that normal samples can be reconstructed well. At inference, a test sample is scored by its reconstruction error, computed as the distance between the original data and its reconstruction version. Samples are flagged as anomalous when it cannot be well explained by the learned representation of "normality", and thus incurs a large reconstruction error. This paradigm is attractive as it requires no labels and offers a simple, unified scoring rule.

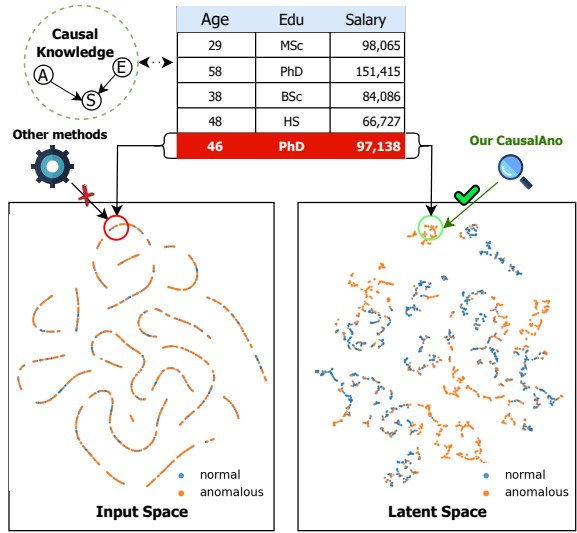

*Figure 1.* A synthetic dataset about income has the causal relationship $Age \rightarrow Salary \leftarrow Edu$, which implies the rule "high Age and high Edu $\rightarrow$ high Salary". Anomalies breaking this relationship (e.g., unusually low salary for given high age and education) are overlapped with normal samples in the input space, making them hard-to-detect for existing AD methods. Our method CausalAno learns causal representations of normal and abnormal samples such that they are clearly separated in the latent space.

However, this reconstruction-and-distance paradigm has two critical limitations. First, the reconstruction error depends

---

[*]Equal contribution   [1]A2I2, Deakin University, Australia [2]Adelaide University, Australia [3]Monash University, Australia. Correspondence to: Dang Nguyen <d.nguyen@deakin.edu.au>.

*Proceedings of the 43rd International Conference on Machine Learning*, Seoul, South Korea. PMLR 306, 2026. Copyright 2026 by the author(s).

on a distance metric (typically, Euclidean distance) that is poorly matched to mixed-type tabular data. When encoded categorical features yield small or poorly calibrated distance changes relative to continuous features, the total reconstruction error is often dominated by continuous dimensions, and violations in discrete fields may be ignored. This creates a blind spot for anomalies that violate structural constraints without producing a large reconstruction distance. For example, a sample "Sex=Male and Pregnancy=Yes" may not appear "far" under reconstruction distances as each attribute is common in isolation even though their joint configuration violates a structural constraint. Second, real-world tabular data often exhibit complex conditional dependencies (Mai et al., 2024; Wang et al., 2024). These complexities demand models that go beyond surface-level statistics to exploit the structural relationships defining how features interact. As illustrated in Figure 1, the dataset has an implicit rule "*high Age and high Edu → high Salary*". Anomalies violating this rule are entangled in the original data space, making them hard-to-detect for existing reconstruction-based methods.

To address these gaps, we propose **CausalAno**, a causal-aware tabular AD method that targets anomalies violating causal relations. It has three steps. First, we estimate a causal graph over the features and train a causal-aware GAN to approximate the structural dependencies that characterize the normal data. Second, since the discriminator is trained under the causal mechanism, we use its penultimate layer as a causality-informed feature extractor to map samples into a latent space. Finally, we fit a Gaussian model on normal embeddings and use the Mahalanobis distance to compute anomaly scores for test samples. Our method is supported by theoretical analysis.

To summarize, we make the following contributions:

(1) We propose CausalAno, the first causal-aware method for unsupervised tabular AD that leverages a causal-aware GAN conditioned on an estimated causal graph to expose anomalies that conflict with learned structural dependencies.

(2) We introduce a unified scoring scheme in the discriminator's latent space, using a Gaussian model with the Mahalanobis distance. Our latent features effectively separate causal mechanism-violating samples from normal samples.

(3) We conduct comprehensive experiments on diverse 28 tabular benchmarks, demonstrating that causal-aware scoring improves detection performance over SOTA statistical, deep learning, and LLM-based methods.

## 2. Related Background

### 2.1. Machine Learning for Tabular Data

Following the success of machine learning (ML) in image and text classification, many ML methods have been adapted to tabular prediction, where structured features are mapped to a target label (Borisov et al., 2022). However, many real-world applications often exhibit severe class imbalance, with long-tailed or rare-event distributions for minority classes (Mai et al., 2024). A common strategy is to increase the size of the minority class by synthesizing additional minority examples to rebalance the training data (Wang et al., 2024).

These needs have motivated a line of work on generative models for tabular data to tackle data scarcity and quality issues. LLM-based generators (Borisov et al., 2023; Nguyen et al., 2024) serialize rows into token sequences and use sequence modeling to capture complex dependencies with flexible conditioning. However, their large size and language-oriented tokenization make them computationally expensive and often poorly aligned with numerical precision and strict tabular constraints (Yang et al., 2025). In contrast, GAN-based models (Xu et al., 2019; Nguyen et al., 2025) learn joint feature distributions via adversarial training and offer a more lightweight and practical choice.

### 2.2. Anomaly Detection

While tabular classification and generation aim to improve predictive performance and data quality, many deployments must also contend with atypical records that can undermine these models in practice. Tabular anomaly detection (AD) aims to identify such atypical rows in *unlabeled* data from high-stakes domains such as finance, cybersecurity, and healthcare. Because explicit anomaly labels are scarce and noisy, most practical methods are *unsupervised* tabular AD (Huang et al., 2025; Kumari et al., 2024).

Most AD methods are correlation-based detectors i.e., they approximate the distribution of normal data and the statistical dependencies among features, then flag deviations as anomalies. Classical methods include proximity-based methods (e.g., KNN, IForest) (Liu et al., 2008; Ramaswamy et al., 2000), distribution-based methods (e.g., PCA, ECOD) (Shyu et al., 2003; Li et al., 2022), and boundary-based one-class methods (e.g., DeepSVDD, GOAD, ICL) (Ruff et al., 2018; Bergman and Hoshen, 2020; Shenkar and Wolf, 2022). LLM-based approaches such as AnoLLM (Tsai et al., 2025) and LLM-DAS (Ye et al., 2026) extend this view by scoring serialized rows with language models or using them to synthesize anomalies. However, in practice, the strongest empirical performance on tabular benchmarks often comes from reconstruction- and generation-based approaches (e.g., RCA, REPEN, NeuTraL, DTE, SLAD) (Pang et al., 2018; Liu et al., 2021; Xu et al., 2023; Qiu et al., 2021; Livernoche et al., 2024). AnoGAN (Schlegl et al., 2019) further optimizes a latent vector per sample and combines its reconstruction error with the discriminator's feature discrepancy. Representation-learning methods (DRL) (Ye et al., 2025) explicitly shape latent space to separate normal and abnor-

mal structure. Across all these families, the central objective remains to approximate the distribution of normal data and the statistical correlations among features. So far, *there is no AD method that explicitly encodes causal mechanisms*.

## 3. The Proposed Framework

### 3.1. Problem Statement

Following previous works (Livernoche et al., 2024; Ye et al., 2025; Tsai et al., 2025), we address AD in an *unsupervised classification setting* i.e., only normal samples are available during training phase. We denote the training set as $\mathcal{D}_{real} = \{x_i\}_{i=1}^N$, where each $x_i \in \mathbb{R}^M$ is a row of a tabular data (i.e., a normal sample) and $N, M$ are the numbers of samples and features. In test phase, a labeled test set is given, $\mathcal{D}_{test} = \{x_i, y_i\}_{i=1}^{N'}$, where $y_i \in \{0, 1\}$ with normals being labeled as 0 while anomalies being labeled as 1.

Our goal is to learn a detector model from $\mathcal{D}_{real}$ such that it constructs a scoring function $\phi : \mathbb{R}^M \to \mathbb{R}$. Given a test sample $x \in \mathcal{D}_{test}$, if $\phi(x)$ is high, it indicates a high probability that $x$ is an anomaly.

### 3.2. Causal Modeling and Anomaly Definitions

We assume that the normal data $\mathcal{D}_{real}$ are generated by a Structural Causal Model (SCM) $\mathcal{M} = \langle \mathcal{X}, \mathcal{E}, \mathcal{F} \rangle$, where $\mathcal{X} = \{X_1, ..., X_M\}$ denotes the features (variables), $\mathcal{E} = \{E_1, \ldots, E_M\}$ represents exogenous noises, and $\mathcal{F} = \{f_1, \ldots, f_M\}$ is a set of deterministic structural functions. Each feature is generated as $X_i = f_i(\mathbf{PA}_i, E_i)$, where $\mathbf{PA}_i$ is the set of causal parents of $X_i$ and $E_i \sim \mathcal{N}(0, 1)$.

Guided by the causal graph $\mathcal{G}_{real}$ derived from $\mathcal{M}$, we construct a *causal-aware generator $G$* composed of $M$ sub-generators $\{G_i\}_{i=1}^M$, where each $G_i$ aims to approximate the corresponding structural function $f_i$. During training, a synthetic example $\hat{x}$ generated by $G$ respects the dependency structure of $\mathcal{G}_{real}$. However, due to the approximation error where $G_i \neq f_i$, the generated sample $\hat{x}$ violates the quantitative causal mechanisms of $\mathcal{M}$. We define such samples as *casual anomalies*.

**Definition 1.** *Causal anomaly*: A synthetic sample $\hat{x}$ generated by $G$ is a causal anomaly if it is produced by a mechanism $G$ that diverges from the true mechanism $\mathcal{F}$ (i.e., $G_i \neq f_i$ for some $i$). For example, given the causal relationship $Age \to Salary \leftarrow Edu$, $\hat{x}$ is a causal anomaly if its Salary is generated via $G_{Salary}([Age, Edu], E) \neq f_{Salary}([Age, Edu], E)$, resulting in a structurally valid but mechanistically impossible value.

In *unsupervised* AD setting, ground-truth anomalies are not available. However, prior works demonstrate that introducing synthetic anomalies during training yields tighter description of normality (Pang et al., 2018; Ruff et al., 2018).

In our work, we leverage the causal-aware generator to provide these anomalies. Crucially, as the sub-generators $G_i$ gradually converge toward $f_i$, the generated causal anomalies evolve from obvious outliers to "hard negatives" that lie closer to the true manifold. This curriculum forces the detector to learn an increasingly refined description of the normal data.

### 3.3. Generator Architecture and Theoretical Justification

We represent the underlying causal structure of the normal data using a Directed Acyclic Graph (DAG) $\mathcal{G}_{real}$, where nodes correspond to the $M$ features and directed edges encode causal dependencies. While the graph skeleton can be estimated using causal discovery algorithms such as PC (Spirtes et al., 2000), these methods do not capture the quantitative mechanisms $\mathcal{F} = \{f_1, \ldots, f_M\}$. Our goal is to learn these mechanisms using a causal-aware generator $G$.

To exploit the structure of $\mathcal{G}_{real}$, we design $G$ as a collection of $M$ structural sub-generators $\{G_i\}_{i=1}^M$. Each $G_i$ models the conditional distribution of variable $X_i$ given its parents $\mathbf{PA}_i$ and an independent noise vector $z_i \sim \mathcal{N}(0, 1)$. Synthetic samples are generated auto-regressively following the topological order of $\mathcal{G}_{real}$ (assumed to be $1, ..., M$ without loss of generality), ensuring that parental values are available before generating each child node.

We provide a theoretical justification for this architecture. We assume the cost function $c$ satisfying the *additive property*: $c([X_A, X_B], [X'_A, X'_B]) = c(X_A, X'_A) + c(X_B, X'_B)$ where $A, B \subset \{1, ..., M\}$ are disjoint with $X_C = [X_i]_{i \in C}$. It is obvious that $c(X_C, X'_C) = \sum_{i \in C} \left| X_i - X'_i \right|^p$ (with $p > 0$) satisfies the additive property.

**Theorem 1.** *Under an additive cost function c, the Wasserstein distance between the normal data distribution $p_\mathcal{D}$ and the generator distribution $p_G$ is lower-bounded by the sum of local structural approximation errors:*

$$\mathcal{W}_c(p_\mathcal{D}, p_G) \geq \sum_{i=1}^M d_i(G_i, f_i), \qquad (1)$$

*where the local structural distances $d_i$ are defined as:*

$$d_i(G_i, f_i) = \\ \min_{\gamma_i \in \Gamma_i} \mathbb{E}_{\gamma_i} \left[ c\left( f_i(X_{\mathbf{PA}_i}, E_i), G_i\left( X_{\mathbf{PA}_i^G}, E_i^G \right) \right) \right], \quad (2)$$

*where $\Gamma_i = \Gamma\left(\alpha_i, \alpha_i^G\right)$ denotes the set of couplings between $\alpha_i$ (the joint distribution of real parents and exogenous noise) and $\alpha_i^G$ (the corresponding distribution induced by the generator).*

Theorem 1 (proof in *Appendix A*) demonstrates that minimizing the global Wasserstein distance $\mathcal{W}_c(p_\mathcal{D}, p_G)$–which

is the objective of our WGAN training–implicitly minimizes the lower bound of local errors. Consequently, this optimization forces each sub-generator $G_i$ to converge towards the true structural function $f_i$, ensuring $G$ captures not just the joint distribution, but the correct causal mechanisms.

### 3.4. Training with Wasserstein GAN

**Adversarial objective:** Guided by Theorem 1, which establishes that minimizing the global Wasserstein distance $\mathcal{W}_c(p_\mathcal{D}, p_G)$ minimizes the local structural approximation error, we adopt the Wasserstein GAN (WGAN) framework (Arjovsky et al., 2017). To tailor this to our problem, we construct the generator $G$ to explicitly mirror the causal graph $\mathcal{G}_{real}$ while training a discriminator $D$ to enforce distributional alignment.

**Discriminator optimization:** We train $D$ using the WGAN-GP objective (Gulrajani et al., 2017) to approximate the Wasserstein distance. The loss function is defined as:

$$\min_{\theta_D} \mathcal{L}_D = -\mathbb{E}_{x \sim p_\mathcal{D}}[D(x)] + \mathbb{E}_{\hat{x} \sim p_G}[D(\hat{x})] + GP, \quad (3)$$

where $p_\mathcal{D}$ and $p_G$ are distributions of normal and synthetic samples, respectively. $D(x)$ and $D(\hat{x})$ denote critic scores. The gradient penalty $GP$ enforces the 1-Lipschitz constraint necessary for the Wasserstein approximation, computed as: $GP = \mathbb{E}_{\tilde{x} \sim p_{\tilde{x}}}[(||\nabla_{\tilde{x}} D(\tilde{x})||_2 - 1)^2]$, where $\tilde{x} = \epsilon x + (1 - \epsilon)\hat{x}$, with $\epsilon \sim \mathcal{U}(0, 1)$.

Crucially, we decompose the discriminator $D$ into a feature extractor $h$ and a linear projection head $w$, such that $D(x) = w^\top h(x)$. This formulation allows us to reinterpret the optimization problem in Equation (3) as:

$$\max_{\theta_h, w} \mathbb{E}_{x \sim p_\mathcal{D}, \hat{x} \sim p_G}[w^\top h(x) - w^\top h(\hat{x})] - GP, \quad (4)$$

where $\theta_h$ are parameters of the feature extractor $h$.

By minimizing $\mathcal{L}_D$, the discriminator learns to map normal samples $x$ and synthetic anomalous samples $\hat{x}$ to distinct regions in the latent space defined by $h$, thereby maximizing their feature-level discrepancy (as illustrated in Figure 1).

**Generator optimization:** Adopting the structural design of CA-GAN (Nguyen et al., 2025), we decompose $G$ into $M$ sub-networks $\{G_i\}_{i=1}^M$. The generation process is autoregressive: given a noise vector $z = [z_1, \ldots, z_M]$ where $z_i \sim \mathcal{N}(0, 1)$, we generate each feature $\hat{X}_i$ according to the topological order of $\mathcal{G}_{real}$:

$$\hat{X}_i = G_i(\mathbf{P\hat{A}}_i, z_i; \theta_G),$$

where $\mathbf{P\hat{A}}_i$ represents the previously generated values of $X_i$'s parents. This ensures that the final synthetic sample $\hat{x} = [\hat{X}_1, ..., \hat{X}_M]$ inherently respects the parent-child dependencies of the causal graph, rather than treating features as an independent vector. Please refer to Figure 2 for the generator's design and workflow.

### 3.5. Generator Optimization and Training Dynamics

To ensure the generator $G$ captures both the data distribution and the underlying causal mechanisms, we optimize it with a dual objective: maximizing the critic score (adversarial goal) while minimizing the structural discrepancy (causal goal). We quantify the latter using the Structural Hamming Distance (SHD) between the causal graph $\mathcal{G}_{real}$ inferred from normal data $\mathcal{D}_{real}$ and $\mathcal{G}_{syn}$ inferred from the generated batch. The total generator loss is defined as:

$$\min_{\theta_G} \mathcal{L}_G = -\underbrace{\mathbb{E}_{\hat{x} \sim p_G}[D(\hat{x})]}_{\text{adversarial loss}} + \lambda \cdot \underbrace{\mathbb{E}_{\hat{x} \sim p_G}[R(\hat{x})]}_{\text{causal loss}}, \quad (5)$$

where $\lambda$ is a coefficient to control the strength of the structural penalty and $R(\hat{x}) = \text{SHD}(\mathcal{G}_{real}, \mathcal{G}_{syn})$. To compute the SHD, we apply the PC algorithm on both $\mathcal{D}_{real}$ to extract $\mathcal{G}_{real}$ and a synthetic batch $\{\hat{x}_i\}_{i=1}^N$ to extract $\mathcal{G}_{syn}$. As the SHD operation involves discrete graph structures and is *non-differentiable* with respect to $\theta_G$, we optimize the causal loss term using the REINFORCE algorithm (Williams, 1992).

While Theorem 1 proves that minimizing the true Wasserstein distance minimizes local structural errors, it assumes the generator strictly explores the constrained couplings $\Gamma_i$. Practically, we use the WGAN-GP objective as a dual approximation, but minimizing it alone does not guarantee structural fidelity. Thus, the SHD penalty acts as an essential soft structural regularizer. It does not enforce exact membership in $\Gamma_i$, but biases the generator toward producing samples whose induced dependency structure aligns with $\mathcal{G}_{real}$, providing an empirical approximation to the theoretical structural constraints.

**Training dynamics:** Following the standard WGAN schedule, we alternate updates between the discriminator $D$ and the generator $G$. This adversarial dynamic creates a natural curriculum for anomaly detection. We illustrate the process in Figure 2. By forcing the discriminator $D$ to distinguish between normal data (valid causal mechanisms) and these early-stage synthetic samples (broken mechanisms), the feature extractor $h$ learns a latent space where normal causal patterns are tightly clustered and separated from anomalies. This learned representation is then directly utilized for anomaly scoring, as detailed in the next section.

### 3.6. Anomaly Scoring via Latent Density Estimation

While the generator $G$ acts as a causal regularizer during training, the feature extractor $h$ (from the discriminator $D$) learns a latent space that maximizes the separability between causally consistent normal samples and mechanism-violating anomalies. We leverage this discriminative manifold for inference.

Unlike reconstruction-based methods (Schlegl et al., 2019; Shyu et al., 2003; Thimonier et al., 2024) which detect

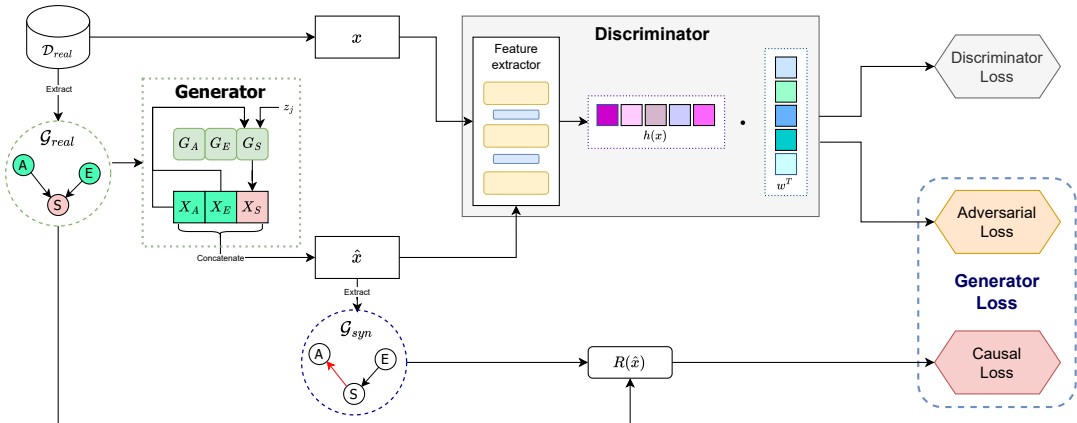

*Figure 2.* **CausalAno**: We train the feature extractor $h$ to maximize the latent separation between normal samples $x \in \mathcal{D}_{real}$ and synthetic anomalies $\hat{x}$ (Equation (4)). To produce these anomalies, we employ a causal generator $G$ composed of structural sub-networks, optimized via joint adversarial and causal losses (Equation (5)). Crucially, we treat early-stage generations as *causal anomalies* i.e., samples that may match marginal distributions but violate the structural dependencies of $\mathcal{D}_{real}$. Finally, CausalAno computes anomaly scores using the Mahalanobis distance in the learned latent space (Equation (6)), flagging samples that deviate from the normal causal manifold.

anomalies based on feature-wise errors in the input space, we quantify normality via *density estimation in the latent feature space*. Given the set of normal embeddings $\boldsymbol{\Phi} = \{h(x_i) \mid x_i \in \mathcal{D}_{real}\}$, we model the normal distribution as a multivariate Gaussian $\mathcal{N}(\mu, \Sigma)$, where the parameters are estimated empirically:

$$\mu = \frac{1}{N}\sum_{i=1}^{N} h(x_i), \Sigma = \frac{1}{N-1}\sum_{i=1}^{N}\big(h(x_i)-\mu\big)\big(h(x_i)-\mu\big)^{\top}$$

This Gaussian assumption is an approximation commonly used in representation-based AD (Rippel et al., 2021; Defard et al., 2021). Moreover, our feature extractor $h$ is trained with the adversarial margin maximization in Equation (4), which empirically encourages the concentration of normal representations in the feature space around a single center $\mu$. This concentration justifies using the Mahalanobis distance as the optimal quadratic test statistic for detecting deviations from the normal cluster (Duda et al., 2001). For a test sample $x' \in \mathcal{D}_{test}$, we compute its anomaly score $s_{x'}$ as:

$$s_{x'} = \big(h(x') - \mu\big)^{\top}\Sigma^{-1}\big(h(x') - \mu\big) \qquad (6)$$

This metric provides a robust anomaly score by explicitly accounting for the covariance between latent features. A high score $s_{x'}$ indicates that $x'$ is likely an anomaly because the sample lies in a low-density region of the latent space, statistically distant from the manifold of causal mechanisms learned during training.

## 4. Experiments

### 4.1. Experiment Setups

**Datasets:** We evaluate our method on a benchmark of 28 tabular anomaly detection datasets, following common prac-

tice in prior works (Livernoche et al., 2024; Tsai et al., 2025; Ye et al., 2025). Specifically, the datasets are collected from *Outlier Detection DataSets* (ODDS) (Rayana, 2016) and *Anomaly Detection Benchmark* (ADBench) (Han et al., 2022), together with several mixed-type datasets from *Kaggle*. Overall, our benchmark contains 10 mixed-type datasets (including categorical and continuous features) and 18 continuous-only datasets. They cover diverse domains (e.g., healthcare, finance, cybersecurity, and social sciences), and span a wide range of characteristics from low-dimensional/small-scale to higher-dimensional/larger-scale settings. Details of 10 mixed-type datasets are in Table 1 while those of 18 continuous datasets are in *Appendix G*.

*Table 1.* Statistics for 10 mixed-type tabular datasets. $M_{cat}$, $M_{con}$, and $N_a$ denote the numbers of categorical features, continuous features, and anomalies. *Lymp* (Lymphography) and *ACD* (Cybersecurity) are from (Rayana, 2016; Capurso, 2024) while *SPD* (Spyware-Attacks), *DAMRE* (Damage-Report), *OS* (OS-Kernel), *SMD* (Smart-Meter) are from Kaggle.

| Dataset | $N$ | $M$ | $M_{cat}$ | $M_{con}$ | $N_a$ |
|---|---|---|---|---|---|
| *Lymp* | 148 | 18 | 15 | 3 | 6 (4%) |
| *ACD* | 10,000 | 4 | 3 | 1 | 490 (5%) |
| *NHIS* | 4,388 | 4 | 2 | 2 | 56 (1%) |
| *SPD* | 1,000 | 12 | 9 | 3 | 764 (76%) |
| *Bank* | 41,188 | 10 | 10 | 0 | 4,640 (11%) |
| *Seismic* | 2,584 | 18 | 4 | 14 | 170 (7%) |
| *CMC* | 1,473 | 8 | 8 | 0 | 29 (2%) |
| *DAMRE* | 1,000 | 5 | 2 | 3 | 100 (10%) |
| *OS* | 1,000 | 5 | 2 | 3 | 91 (9%) |
| *SMD* | 5,000 | 7 | 2 | 5 | 250 (5%) |

**Evaluation metrics:** To ensure our evaluation is directly comparable to established tabular anomaly detection studies,

we follow the same train–test construction protocol used in prior works (Zong et al., 2018; Bergman and Hoshen, 2020; Livernoche et al., 2024; Tsai et al., 2025; Ye et al., 2025). We build the training set by randomly sampling 50% of the normal data, and form the test set by combining the remaining 50% normal samples with all anomaly samples. In line with benchmark reporting conventions in the literature (Han et al., 2022; Tsai et al., 2025; Ye et al., 2025), we report AUC-ROC as our primary evaluation metric as it measures ranking quality between normal and anomalous instances across decision thresholds. We also report other metrics such as F1-score and ranking in *Appendix* C.

**Baseline methods:** We compare against a diverse suite of 16 SOTA baselines, which covers from classical detectors, deep representation learning methods, self-supervised methods, and generative models to recent LLM-assisted methods. They are *IForest* (Liu et al., 2008), *KNN* (Ramaswamy et al., 2000), *PCA* (Shyu et al., 2003), *ECOD* (Li et al., 2022), *DeepSVDD* (Ruff et al., 2018), *REPEN* (Pang et al., 2018), *RCA* (Liu et al., 2021), *SLAD* (Xu et al., 2023), *GOAD* (Bergman and Hoshen, 2020), *NeuTraL* (Qiu et al., 2021), *ICL* (Shenkar and Wolf, 2022), *DTE* (Livernoche et al., 2024), *AnoGAN* (Schlegl et al., 2019), *DRL* (Ye et al., 2025), *AnoLLM* (Tsai et al., 2025), and *LLM-DAS* (Ye et al., 2026).

To ensure fair comparison and reproducibility, we run baselines using publicly available implementations: 13 methods from IForest to AnoGAN are executed via the PyOD ecosystem while AnoLLM[1], DRL[2], and LLM-DAS[3] are run using the authors' released codebases. All baselines follow the same data splits, preprocessing pipeline, and evaluation metric as our method CausalAno.

**Implement details:** Following the CA-GAN implementation (Nguyen et al., 2025), we use the same network architecture and hyper-parameter settings for the discriminator $D$ and causal generator $G$. The discriminator is a 3-layer MLP with two hidden layers of 256 units acting as the feature extractor $h$. The generator is decomposed into sub-generators $\{G_i\}$, each of them is a 4-layer MLP. The activation function for the generator's output layer is type-specific: $\tanh$ for continuous variables and Gumbel-Softmax for categorical variables. We run the PC algorithm with $\alpha = 0.05$ to extract causal graphs. More details are in *Appendix* B.

To strictly adhere to the unsupervised setting and prevent label leakage from the test set, we avoided dataset-specific hyper-parameter tuning. Instead, we adopt a **fixed configuration** across all datasets, following CA-GAN (Nguyen et al., 2025). Namely, we set epochs $= 300$, noise dimen-

sion $= 16$, and causal coefficient $\lambda = 0.01$. The batch size is set to 64 to balance gradient stability with training speed. This fixed-parameter protocol demonstrates CausalAno's robustness without requiring extensive tuning for each new domain. We provide a sensitivity analysis of $\lambda$ in *Appendix* D, showing that performance remains stable within the range $[0.01, 0.1]$. To mitigate randomness, we repeat each experiment five times with different random seeds and report the average performance.

### 4.2. Results and Analysis

#### 4.2.1. RESULTS ON MIXED-TYPE DATASETS

Table 2 summarizes AUC-ROC (%) of all methods on the 10 mixed-type datasets. In general, our CausalAno achieves the best average score (76.76), outperforming the strongest baseline DRL (73.06) by 3%. CausalAno also attains the top-2 result on 9 out of 10 datasets, indicating that its excellent performance are not driven by a single dataset but generalize across heterogeneous mixed-type datasets.

We attribute the consistent gains of CausalAno on mixed-type datasets to two key design choices. First, it leverages a causal-aware discriminator trained under the causal factorization, making the representation more sensitive to the violations of learned causal mechanisms rather than the surface-level feature irregularities. Second, the anomaly score computed via the Mahalanobis distance captures joint deviations with covariance normalization, which tends to be more robust on mixed-type data where categorical and continuous features may cause overlap or dominance effects under simpler L1 and cosine distances (Schlegl et al., 2019; Ye et al., 2025).

#### 4.2.2. RESULTS ON CONTINUOUS DATASETS

Figure 3 presents the average AUC-ROC performance over the 18 continuous-only datasets (Rayana, 2016; Han et al., 2022). Our method CausalAno is the best method, following by KNN and DRL. It achieves 0.93 AUC-ROC, surpassing 2-3% over the strongest competing baselines KNN and DRL, which are widely recognized as top performers on numerical-only tabular AD tasks. Since these benchmarks contain only numerical columns, they reduce the potential advantage of LLM-based methods that rely on rich column semantics.

The results in Table 2 and Figure 3 indicate that CausalAno retains strong robustness on numerical-only benchmarks while delivering larger gains on mixed-type datasets where modeling structural dependencies is more critical.

---

[1]https://github.com/amazon-science/AnoLLM-large-language-models-for-tabular-anomaly-detection

[2]https://github.com/HangtingYe/DRL

[3]https://github.com/HangtingYe/LLM_DAS

*Table 2.* AUC-ROC scores (%) for all methods on 10 mixed-type datasets. **Bold** and underline indicate the best and second-best methods. We report standard deviation in *Appendix* C.

| | Lymp | ACD | NHIS | SPD | Bank | Seismic | CMC | DAMRE | OS | SMD | AVG |
|---|---|---|---|---|---|---|---|---|---|---|---|
| Iforest | 72.44 | 50.59 | 66.45 | 50.82 | 51.17 | 69.73 | 51.98 | 66.28 | 78.25 | 81.32 | 63.90 |
| KNN | 88.19 | 49.31 | 67.43 | 49.14 | 51.09 | **74.21** | 56.77 | 92.69 | 92.71 | 92.38 | 71.39 |
| PCA | 85.87 | 50.53 | 65.75 | 49.15 | 51.45 | 70.18 | 50.94 | 87.90 | 84.80 | 93.26 | 68.98 |
| ECOD | 84.32 | 50.70 | 66.06 | 49.47 | 51.25 | 69.70 | 53.60 | 83.91 | 76.33 | 91.08 | 67.64 |
| DeepSVDD | 88.50 | 48.00 | 67.30 | 48.55 | 50.18 | 68.99 | 57.01 | 91.17 | 79.91 | 67.50 | 66.71 |
| GOAD | 83.15 | 49.95 | 69.90 | 49.13 | 51.27 | 71.71 | 47.78 | 88.50 | 90.40 | 14.73 | 61.65 |
| ICL | 92.49 | **53.39** | 77.03 | 48.64 | 52.42 | 71.51 | 54.42 | 92.21 | **95.81** | 85.05 | 72.30 |
| RCA | 91.78 | 50.55 | 66.80 | 49.53 | 51.23 | 72.89 | 50.52 | 91.33 | 86.38 | 92.72 | 70.37 |
| SLAD | 93.94 | 50.41 | 82.43 | 46.11 | 53.02 | 70.88 | 49.59 | 77.66 | 86.50 | 47.93 | 65.85 |
| NeuTral | 76.01 | 49.53 | 66.73 | **52.73** | 50.92 | 67.44 | 54.03 | 82.83 | 79.66 | 51.66 | 63.15 |
| DTE | 86.66 | 51.24 | 48.69 | 48.38 | 58.71 | 70.15 | 54.01 | 88.40 | 93.86 | 80.57 | 68.07 |
| REPEN | 84.60 | 50.28 | 67.36 | 47.89 | 51.07 | 72.45 | 57.06 | 77.80 | 91.71 | 80.18 | 68.04 |
| AnoGAN | 98.69 | 48.84 | 47.90 | 50.77 | 51.56 | 70.85 | 46.71 | 67.89 | 67.35 | 72.22 | 62.28 |
| DRL | 94.46 | 49.56 | **88.88** | 49.49 | 56.66 | 67.41 | 57.46 | 88.64 | 94.08 | 83.93 | 73.06 |
| AnoLLM | **99.63** | 50.91 | 53.37 | 47.89 | 64.63 | 73.93 | 53.68 | **93.93** | 92.03 | 91.57 | 72.16 |
| LLM-DAS | 99.44 | 50.24 | 77.57 | 49.48 | 61.69 | 70.03 | 50.83 | 89.68 | 85.42 | **95.74** | 73.01 |
| CausalAno | 98.31 | 51.93 | 84.66 | 51.84 | **67.67** | 73.75 | **57.89** | 93.11 | 94.71 | 93.69 | **76.76** |

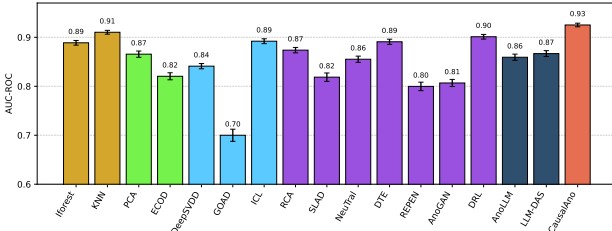

*Figure 3.* Average AUC-ROC scores with standard deviation for all methods over 18 continuous datasets. The color scheme is: yellow (proximity-based), green (distribution-based), blue (boundary-based), purple (reconstruction- and generation-based), navy (LLM-based), orange (ours).

tural dependency violations–a key characteristic of causal anomalies in tabular data.

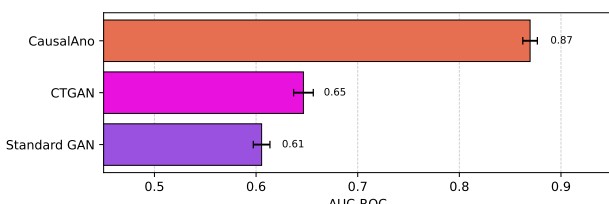

*Figure 4.* Effect of the causal GAN backbone on AD performance on all 28 datasets: AUC-ROC comparison between CausalAno and standard GAN and CTGAN.

### 4.3. Ablation Study

#### 4.3.1. CAUSAL GAN VS. OTHER GAN BACKBONES

To quantify the effectiveness of using a causal GAN, we conduct an ablation to isolate its impact. Namely, we keep the overall CausalAno pipeline unchanged, and only replace the causal GAN component with standard GAN (Goodfellow et al., 2014) and CTGAN (Xu et al., 2019).

Figure 4 shows the average AUC-ROC of each alternative on all 28 datatsets. By substituting our causal GAN with other GANs, it leads to a clear performance drop: the CTGAN decreases to 0.65 while the standard GAN further decreases to 0.61. In contrast, our CausalAno achieves average 0.87 AUC-ROC on all 28 datasets. This highlights that the gain of CausalAno benefits from the causal factorization in the causal GAN, which encourages the discriminator to learn causal-aware representations that are more sensitive to struc-

#### 4.3.2. VISUALIZATION

We visualize the t-SNE projections (Van der Maaten and Hinton, 2008) of latent representations learned by AnoGAN (GAN-based method) and DRL (the best baseline) on four datasets Annthyroid, OS, DAMRE, and Pendigits. Our goal is to examine how different methods separate normal and anomalous samples in the latent space.

As shown in Figure 5(a)-(c) where the normals and anomalies are entangled in the original observed space, the normal and anomalous embeddings learned by the baselines remain interleaved or collapse in the latent space. Only CausalAno maps anomalies into more localized regions that are better separated from the dominant normal mass. Figure 5(d) provides a complementary case where the input space already shows relatively clear normal-abnormal structure, but competing methods compress the data and blur this distinction in the latent space, leading to increased overlap. In contrast,

CausalAno better preserves the separation, consistent with its stronger detection results in Table 4. This example is interpreted as evidence of representation stability rather than visual separation improvement.

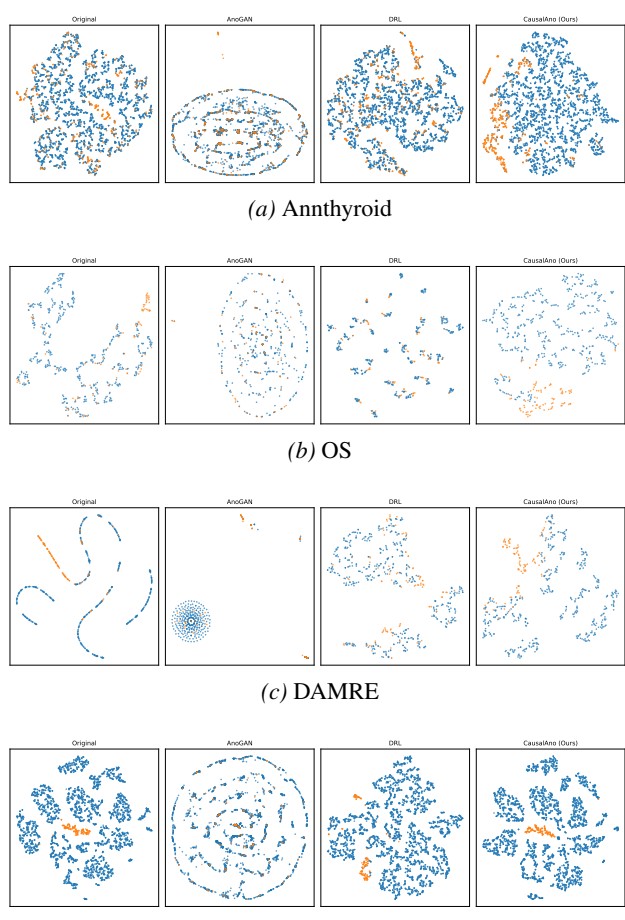

*(a)* Annthyroid

*(b)* OS

*(c)* DAMRE

*(d)* Pendigits

*Figure 5.* t-SNE visualization for original data space and models' latent space on four real-world datasets. Blue and orange points denote normal and anomalous samples. Only CausalAno produces a clearly separated latent space between normals and anomalies.

### 4.3.3. CAUSAL ANOMALY DETECTION

To validate CausalAno's ability to detect causal anomalies that violate causal knowledge, we use a controlled synthetic dataset with three variables–Age (continuous), Edu (categorical), and Salary (continuous)–under the assumed causal structure $Age \rightarrow Salary \leftarrow Edu$. We generate $\mathcal{D}_{real}$ (10,000 normals) by sampling $Age \sim \mathcal{U}(22, 65)$ and $Edu \in \{0 : \text{HighSchool}, 1 : \text{Bachelors}, 2 : \text{Masters}, 3 : \text{PhD}\}$ with probabilities $[0.3, 0.4, 0.2, 0.1]$, then computing Salary using the causal mechanism $30000 + (Age \times 1000) + (Edu \times 20000) + E$, where $E \sim \mathcal{N}(0, 5000)$ is a noise. For $\mathcal{D}_{test}$ (2,000 samples), we mix 1,000 normals (same causal mechanism) and 1,000 causal anomalies created by shuffling the parent values (Age and/or Edu) when computing Salary, pro-

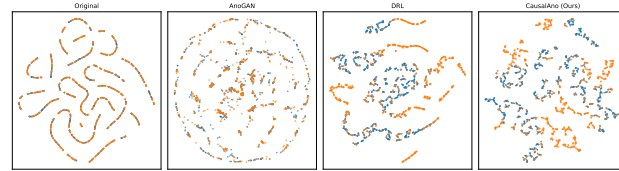

*Figure 6.* t-SNE visualization for original data space and models' latent space on the Synthetic dataset.

ducing mismatched tuples $(Age, Edu, Salary)$ that break the causal rule.

Quantitatively, CausalAno detects these causal anomalies most reliably (813/1000), outperforming DRL (710/1000) and AnoGAN (508/1000). Figure 6 visualizes the learned embeddings, where AnoGAN appears partly separable but unreliable as far fewer anomalous points are clearly visible. DRL remains heavily entangled whereas only CausalAno forms more compact, better-separated clusters, consistent with its higher true positive rate.

Table 3 further lists five representative normal and abnormal records, where normal samples follow the expected rule "high Age and high Edu $\rightarrow$ high Salary" while anomalies exhibit clear parent-child mismatches. For example, the first record (Age=36, Edu=PhD, Salary=97,138) is unusually low whereas the second record (Age=31, Edu=HighSchool, Salary=116,616) is unexpectedly high for the given context. Our CausalAno correctly detects all five causal anomalies, where it identifies Salary as the primary violating component relative to its parents (Age, Edu).

*Table 3.* Example normal and abnormal samples.

| Normal samples | | | Abnormal samples | | |
|---|---|---|---|---|---|
| Age | Edu | Salary | Age | Edu | Salary |
| 29 | MSc | 98,065 | 36 | PhD | 97,138 |
| 58 | PhD | 151,415 | 31 | HS | 116,616 |
| 38 | BSc | 84,086 | 57 | HS | 101,899 |
| 48 | HS | 66,727 | 62 | HS | 77,970 |
| 56 | HS | 86,920 | 41 | BSc | 101,004 |

### 4.3.4. CONTAMINATED TRAINING DATA

Real-world AD often involves mildly contaminated training data. To evaluate robustness, we conduct experiments on four datasets (Annthyroid, OS, DAMRE, and SMD) with contamination ratios ranging from 0% to 5%. As shown in Figure 7, CausalAno maintains stable performance across all contamination levels and consistently outperforms DRL. As anomaly scoring is performed in the learned latent space, the influence of a small number of contaminated points is limited when estimating the mean and covariance.

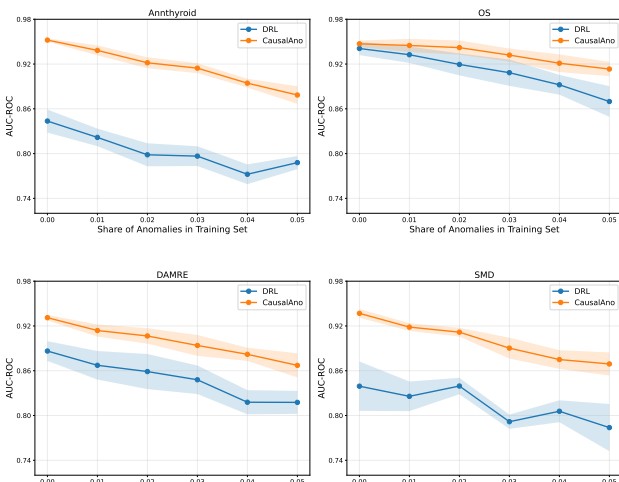

*Figure 7.* AUC-ROC of CausalAno with different contamination ratios in training data.

#### 4.3.5. ADDITIONAL EXPERIMENTS

**Computational efficiency:** *Appendix* E compares the runtime of CausalAno with other methods.

**Robustness to causal graph** $\mathcal{G}_{real}$**:** *Appendix* F shows how sensitive of CausalAno to the correctness of $\mathcal{G}_{real}$.

## 5. Conclusion

We address unsupervised AD in tabular data where anomalies often arise as violations of underlying structural dependencies rather than simple marginal outliers. Different from correlation-driven detectors, our method CausalAno explicitly incorporates a causal structure to expose mechanism-inconsistent samples. First, we train a discriminator to learn causal-aware latent representations. Second, we fit a Gaussian model in the discriminator's latent space and use the Mahalanobis distance to separate causal anomalies from normal samples. Finally, extensive experiments on 28 datasets show that CausalAno achieves strong and consistent performance while delivering larger gains on mixed-type datasets where modeling structural dependencies is more critical.

Regarding future work, handling raw free-text attributes in tabular data remains an important challenge for AD tasks. This typically requires additional representation learning, such as pretrained language model embeddings (Tsai et al., 2025). In principle, such embeddings could be utilized as inputs to CausalAno, and we plan to investigate this extension. Furthermore, we intend to apply CausalAno to the mental health domain (Newby et al., 2025), where unsupervised AD is increasingly utilized to flag sudden, atypical shifts in patient behavioral metrics, providing preemptive clinical alerts for acute psychological distress (Henson et al., 2021).

## Acknowledgment

This research was partially supported by the Wellcome Trust [grant number: 303030/Z/23/Z].

## Impact Statement

This work aims to advance Machine Learning by introducing a causal-aware framework for Tabular Anomaly Detection. By leveraging causal mechanisms to identify anomalies, this research seeks to improve the robustness and reliability of automated systems in high-stakes domains, such as healthcare monitoring and financial security. While the method is general-purpose, we believe that incorporating causal grounding can mitigate risks associated with spurious correlations, thereby enhancing the stability of anomaly detection in complex, real-world deployments.

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

# A. Proof of Theorem 1:

We depart from the last feature $X_M$. Let $\beta_{1:M}$ be a joint distribution over $(X_1, ..., X_M)$ and $(X_1^G, ..., X_M^G)$ where $X_{1:M}$ is sampled from the ground-truth SCM and $X_{1:M}^G$ is sampled from the SCM induced by the causal-aware generator. We have:

$$\mathbb{E}_{\beta_{1:M}}\left[c\left(X_{1:M}, X_{1:M}^G\right)\right] = \mathbb{E}_{\beta_{1:M}}\left[c\left(X_{1:M-1}, X_{1:M-1}^G\right) + c\left(X_M, X_M^G\right)\right]$$
$$= \mathbb{E}_{\beta_{1:M-1}}\left[c\left(X_{1:M-1}, X_{1:M-1}^G\right)\right] + \mathbb{E}_{(X_M, X_M^G)\sim\beta_M}\left[c\left(X_M, X_M^G\right)\right]$$

where $\beta_{1:M-1}$ and $\beta_M$ are the projections of $\beta_{1:M}$ onto $X_{1:M-1}, X_{1:M-1}^G$ and $X_M, X_M^G$ respectively.

We now focus on the second term. We consider three distributions: (1) $\beta_M$ over $X_M, X_M^G$, (2) $\delta_M$ over $[X_{\mathbf{PA}_M}, E_M]$ and $f_M(X_{\mathbf{PA}_M}, E_M)$, and (3) $\delta_M^G$ over $[X_{\mathbf{PA}_M^G}, E_M^G]$ and $G_M(X_{\mathbf{PA}_M^G}, E_M^G)$.

Using the gluing theorem (see Lemma 5.5 in (Santambrogio, 2015)), there exists a distribution $\rho$ over $[X_{\mathbf{PA}_M}, E_M]$, $f_M(X_{\mathbf{PA}_M}, E_M) = X_1$, $[X_{\mathbf{PA}_M^G}, E_M^G]$, and $G_M(X_{\mathbf{PA}_M^G}, E_M^G) = X_M^G$ such that the projections onto two corresponding variables reduce to $\beta_M, \delta_M$, and $\delta_M^G$ respectively. We denote $\gamma_M \in \Gamma_M$ as the projection of $\rho$ onto $[X_{\mathbf{PA}_M}, E_M]$ and $[X_{\mathbf{PA}_M^G}, E_M^G]$. From the definition of the chain of distributions in (1), (2), and (3), we have $(f_M, G_M)\#\gamma_M = \beta_M$.

The second term can be rewritten as:

$$\mathbb{E}_{(X_M, X_M^G)\sim\beta_M}\left[c\left(X_M, X_M^G\right)\right] = \mathbb{E}_{\left(\left[X_{\mathbf{PA}_M}, E_M\right], \left[X_{\mathbf{PA}_M^G}, E_M^G\right]\right)\sim\gamma_M}\left[c\left(f_M\left([X_{\mathbf{PA}_M}, E_M]\right), G_M\left(X_{\mathbf{PA}_M^G}, E_M^G\right)\right)\right]$$

It follows that:

$$W_c(p_{\mathcal{D}}, p_G) = \min_{\beta_{1:M}}\mathbb{E}_{\beta_{1:M}}\left[c\left(X_{1:M}, X_{1:M}^G\right)\right]$$
$$\geq \min_{\beta_{1:M-1}}\mathbb{E}_{\beta_{1:M-1}}\left[c\left(X_{1:M-1}, X_{1:M-1}^G\right)\right] +$$
$$\min_{\gamma_M\in\Gamma_M}\mathbb{E}_{\left(\left[X_{\mathbf{PA}_M}, E_M\right], \left[X_{\mathbf{PA}_M^G}, E_M^G\right]\right)\sim\gamma_M}\left[c\left(f_M([X_{\mathbf{PA}_M}, E_M]), G_M\left(X_{\mathbf{PA}_M^G}, E_M^G\right)\right)\right]$$
$$= \min_{\beta_{1:M-1}}\mathbb{E}_{\beta_{1:M-1}}\left[c\left(X_{1:M-1}, X_{1:M-1}^G\right)\right] + d_M(f_M, G_M)$$

By applying the same derivations backward recursively for $M-1, M-2, ..., 1$, we reach the conclusion.

# B. Network Architecture

## B.1. Sub-generator $G_i$

Each sub-generator $G_i$ for a variable $X_i$ is implemented as a four-layer multilayer perceptron (MLP) that receives as its input the concatenation of the parent variables $\mathbf{PA}_i$ and a latent noise vector $z_i \sim \mathcal{N}(0, 1)$. The network applies the LeakyReLU activations with a negative slope of 0.2 and Batch Normalization with momentum of 0.8 after the second and third layers to enhance stability during training. The hidden layers progressively expand from 64 to 128 units, maintaining consistent activation and normalization across layers. The output layer depends on the variable type: for a continuous variable, a *tanh activation* produces the final value whereas for a categorical variable, a *Gumbel-Softmax layer* outputs a differentiable probability vector over $K$ categories.

## B.2. Discriminator $D$

The discriminator is designed as a three-layer MLP that maps a complete record $x$ to a scalar authenticity score $D(x)$. The first two hidden layers use LeakyReLU activations with a slope of 0.2, each of them contains 256 units, followed by a linear output layer producing a single scalar. Both the generator and the discriminator are optimized using the Adam optimizer with a learning rate of $2 \times 10^{-4}$, $\beta_1 = 0.5$, and $\beta_2 = 0.9$.

## C. Full Performance

### C.1. AUC-ROC

We report AUC-ROC as the AD metric in the main paper. Figure 8 summarizes the average AUC-ROC across all 28 datasets while Table 4 reports AUC-ROC for each dataset.

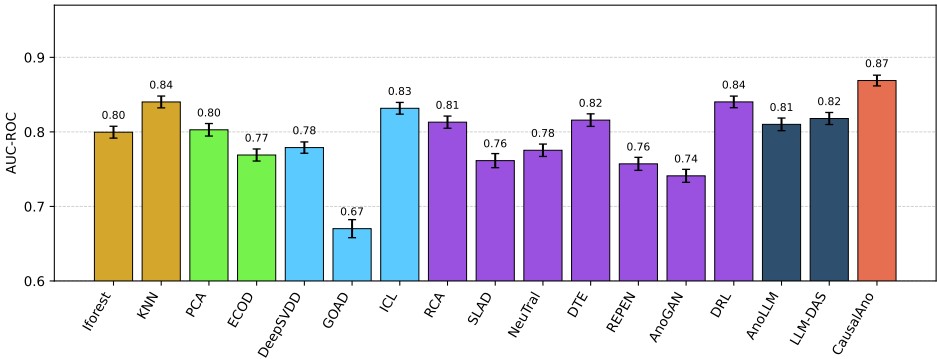

*Figure 8.* Average AUC-ROC scores with standard deviation bars for all methods over all 28 datasets (*higher is better*).

*Table 4.* AUC-ROC (standard deviation) on each dataset (*higher is better*).

| Dataset | Iforest | PCA | KNN | ECOD | DeepSVDD | RCA | SLAD | GOAD | Neutral | ICL | DTE | REPEN | AnoGAN | AnoLLM | DRL | LLM-DAS | CausalAno |
|---|---|---|---|---|---|---|---|---|---|---|---|---|---|---|---|---|---|
| Annthyroid | 0.9058 | 0.8386 | 0.8112 | 0.7898 | 0.7332 | 0.7175 | 0.7512 | 0.5485 | 0.7815 | 0.8323 | **0.9741** | 0.7154 | 0.6495 | 0.9165 | 0.8436 | 0.8281 | 0.9521 |
|  | (0.009) | (0.030) | (0.004) | (0.003) | (0.011) | (0.003) | (0.007) | (0.009) | (0.018) | (0.022) | (0.004) | (0.024) | (0.127) | (0.002) | (0.021) | (0.029) | (0.003) |
| BreastW | **0.9929** | 0.9888 | 0.9916 | 0.9893 | 0.9712 | 0.9866 | 0.9880 | 0.9924 | 0.9843 | 0.9896 | 0.9836 | 0.9588 | 0.9625 | 0.9915 | 0.9893 | 0.9873 | 0.9877 |
|  | (0.002) | (0.005) | (0.003) | (0.002) | (0.003) | (0.004) | (0.006) | (0.002) | (0.004) | (0.004) | (0.007) | (0.018) | (0.058) | (0.002) | (0.003) | (0.005) | (0.003) |
| Ecoli | 0.8257 | 0.8365 | 0.8786 | 0.7508 | 0.8510 | 0.8821 | 0.7968 | 0.8169 | 0.8638 | **0.8931** | 0.8276 | 0.8496 | 0.7735 | 0.8557 | 0.8522 | 0.8332 | 0.8874 |
|  | (0.008) | (0.008) | (0.002) | (0.012) | (0.015) | (0.024) | (0.027) | (0.007) | (0.006) | (0.013) | (0.013) | (0.041) | (0.030) | (0.022) | (0.022) | (0.009) | (0.008) |
| Mammography | 0.8788 | 0.8980 | 0.8709 | **0.9062** | 0.8695 | 0.8755 | 0.7435 | 0.7592 | 0.6681 | 0.7901 | 0.8501 | 0.8508 | 0.8047 | 0.8565 | 0.8435 | 0.9046 | 0.8257 |
|  | (0.004) | (0.004) | (0.001) | (0.001) | (0.009) | (0.004) | (0.011) | (0.005) | (0.0560) | (0.015) | (0.019) | (0.011) | (0.122) | (0.002) | (0.013) | (0.003) | (0.020) |
| Shuttle | 0.9965 | 0.9934 | 0.9990 | 0.9929 | 0.9900 | 0.9955 | 0.9982 | 0.9882 | 0.9980 | 0.9994 | 0.9978 | 0.9945 | 0.9892 | 0.9984 | 0.9992 | 0.9954 | **0.9998** |
|  | (0.001) | (0.001) | (0.000) | (0.000) | (0.008) | (0.000) | (0.000) | (0.002) | (0.002) | (0.001) | (0.000) | (0.002) | (0.004) | (0.000) | (0.001) | (0.000) | (0.000) |
| Thyroid | 0.9890 | 0.9835 | 0.9760 | 0.9787 | 0.8617 | 0.9715 | 0.9493 | 0.6895 | 0.8389 | 0.9779 | **0.9910** | 0.9330 | 0.9598 | 0.9891 | 0.9808 | 0.9759 | 0.9858 |
|  | (0.001) | (0.002) | (0.002) | (0.001) | (0.021) | (0.002) | (0.010) | (0.023) | (0.017) | (0.010) | (0.001) | (0.014) | (0.016) | (0.000) | (0.007) | (0.003) | (0.002) |
| Wine | 0.9672 | 0.9949 | 0.9891 | 0.7413 | 0.9637 | 0.9451 | 0.9947 | **0.9972** | 0.9941 | 0.9787 | 0.9941 | 0.2746 | 0.9802 | 0.9960 | 0.9963 | 0.9874 | 0.9961 |
|  | (0.011) | (0.004) | (0.005) | (0.030) | (0.020) | (0.014) | (0.010) | (0.002) | (0.001) | (0.015) | (0.005) | (0.068) | (0.011) | (0.003) | (0.004) | (0.011) | (0.003) |
| Waveform | 0.7126 | 0.6553 | 0.7606 | 0.6043 | 0.5805 | 0.6548 | 0.4854 | 0.4767 | 0.7343 | 0.6861 | 0.6283 | 0.7588 | 0.6064 | 0.6496 | 0.6638 | 0.6815 | **0.7876** |
|  | (0.016) | (0.004) | (0.008) | (0.003) | (0.018) | (0.006) | (0.009) | (0.010) | (0.008) | (0.032) | (0.025) | (0.053) | (0.119) | (0.008) | (0.040) | (0.006) | (0.011) |
| WBC | **0.9946** | 0.9933 | 0.9916 | 0.9933 | 0.9088 | 0.9847 | 0.9914 | 0.9497 | 0.8540 | 0.9774 | 0.9226 | 0.9858 | 0.9944 | **0.9946** | 0.9727 | 0.9918 | 0.9778 |
|  | (0.002) | (0.001) | (0.002) | (0.002) | (0.027) | (0.004) | (0.003) | (0.029) | (0.062) | (0.010) | (0.049) | (0.004) | (0.002) | (0.002) | (0.008) | (0.003) | (0.008) |
| Pendigits | 0.9689 | 0.9426 | **0.9986** | 0.9279 | 0.8227 | 0.9698 | 0.9307 | 0.2551 | 0.9652 | 0.9743 | 0.9775 | 0.9295 | 0.9443 | 0.9512 | 0.9810 | 0.9449 | 0.9968 |
|  | (0.006) | (0.002) | (0.000) | (0.002) | (0.055) | (0.003) | (0.012) | (0.023) | (0.007) | (0.024) | (0.010) | (0.016) | (0.002) | (0.002) | (0.006) | (0.005) | (0.003) |
| Cardio | 0.9474 | 0.9653 | 0.9181 | 0.9335 | 0.8437 | 0.9499 | 0.8392 | 0.4897 | 0.8353 | 0.8739 | 0.9133 | 0.8559 | 0.9046 | 0.8942 | 0.9070 | **0.9679** | 0.9403 |
|  | (0.005) | (0.001) | (0.004) | (0.003) | (0.019) | (0.003) | (0.022) | (0.029) | (0.010) | (0.030) | (0.018) | (0.047) | (0.040) | (0.013) | (0.025) | (0.005) | (0.010) |
| PageBlocks | 0.9331 | 0.9325 | 0.9473 | 0.9135 | 0.9016 | 0.9385 | 0.8560 | 0.8718 | 0.9139 | 0.9398 | 0.9524 | 0.8866 | 0.8628 | 0.8817 | 0.9666 | 0.9304 | **0.9722** |
|  | (0.002) | (0.002) | (0.001) | (0.002) | (0.012) | (0.001) | (0.018) | (0.007) | (0.003) | (0.005) | (0.002) | (0.004) | (0.027) | (0.008) | (0.003) | (0.002) | (0.003) |
| Stamps | 0.9428 | 0.9367 | 0.9319 | 0.8870 | 0.8475 | 0.9328 | 0.6727 | 0.5529 | 0.8219 | 0.8784 | 0.8786 | 0.8865 | 0.7411 | 0.9359 | 0.9305 | 0.9251 | **0.9503** |
|  | (0.008) | (0.009) | (0.012) | (0.015) | (0.038) | (0.013) | (0.040) | (0.046) | (0.034) | (0.046) | (0.044) | (0.053) | (0.255) | (0.008) | (0.016) | (0.012) | (0.016) |
| Glass | 0.7857 | 0.7040 | 0.8436 | 0.6841 | 0.8522 | 0.7176 | 0.7933 | 0.6615 | **0.9236** | 0.9191 | 0.7767 | 0.7301 | 0.6272 | 0.7655 | 0.8703 | 0.6902 | 0.8843 |
|  | (0.014) | (0.013) | (0.018) | (0.016) | (0.035) | (0.024) | (0.032) | (0.044) | (0.022) | (0.034) | (0.026) | (0.036) | (0.163) | (0.016) | (0.038) | (0.014) | (0.033) |
| Vowels | 0.7711 | 0.6363 | 0.9732 | 0.5958 | 0.8948 | 0.8929 | 0.9715 | 0.9167 | **0.9919** | 0.9816 | 0.9778 | 0.7398 | 0.5841 | 0.6559 | 0.9864 | 0.6732 | 0.9902 |
|  | (0.017) | (0.010) | (0.002) | (0.005) | (0.019) | (0.003) | (0.003) | (0.019) | (0.002) | (0.003) | (0.010) | (0.066) | (0.054) | (0.004) | (0.006) | (0.011) | (0.003) |
| MagicGamma | 0.7743 | 0.7065 | 0.8453 | 0.6397 | 0.7179 | 0.7566 | 0.7412 | 0.6518 | 0.8214 | 0.7927 | **0.8485** | 0.7640 | 0.6839 | 0.7266 | 0.8001 | 0.7114 | 0.8368 |
|  | (0.005) | (0.002) | (0.002) | (0.002) | (0.0098) | (0.002) | (0.004) | (0.007) | (0.011) | (0.015) | (0.012) | (0.020) | (0.036) | (0.001) | (0.012) | (0.002) | (0.012) |
| Yeast | 0.8532 | **0.8568** | 0.8000 | 0.7907 | 0.6576 | 0.8166 | 0.4053 | 0.1744 | 0.5131 | 0.6921 | 0.7295 | 0.7107 | 0.7585 | 0.7362 | 0.7615 | 0.8375 | 0.7925 |
|  | (0.007) | (0.008) | (0.006) | (0.007) | (0.036) | (0.008) | (0.060) | (0.014) | (0.043) | (0.037) | (0.074) | (0.118) | (0.138) | (0.009) | (0.018) | (0.009) | (0.009) |
| ImgSeg | 0.7582 | 0.7172 | 0.8585 | 0.6484 | 0.8715 | 0.7399 | 0.8270 | 0.8072 | **0.8909** | 0.8798 | 0.8119 | 0.5707 | 0.6945 | 0.6721 | 0.8754 | 0.7355 | 0.8895 |
|  | (0.013) | (0.008) | (0.007) | (0.006) | (0.007) | (0.009) | (0.009) | (0.009) | (0.009) | (0.024) | (0.015) | (0.033) | (0.067) | (0.015) | (0.015) | (0.016) | (0.011) |
| Lymphography | 0.7244 | 0.8587 | 0.8819 | 0.8432 | 0.8850 | 0.9178 | 0.9394 | 0.8315 | 0.7601 | 0.9249 | 0.8666 | 0.8460 | 0.9869 | **0.9963** | 0.9446 | 0.9944 | 0.9831 |
|  | (0.079) | (0.029) | (0.041) | (0.030) | (0.040) | (0.022) | (0.024) | (0.065) | (0.039) | (0.023) | (0.041) | (0.038) | (0.009) | (0.004) | (0.036) | (0.006) | (0.010) |
| Cybersecurity (ACD) | 0.5059 | 0.5053 | 0.4931 | 0.5070 | 0.4800 | 0.5055 | 0.5041 | 0.4995 | 0.4953 | **0.5339** | 0.5124 | 0.5028 | 0.4884 | 0.5091 | 0.4956 | 0.5024 | 0.5193 |
|  | (0.010) | (0.013) | (0.007) | (0.012) | (0.005) | (0.012) | (0.008) | (0.012) | (0.019) | (0.005) | (0.008) | (0.008) | (0.021) | (0.005) | (0.008) | (0.008) | (0.008) |
| NHIS-Claims | 0.6645 | 0.6575 | 0.6743 | 0.6606 | 0.6730 | 0.6680 | 0.8243 | 0.6990 | 0.6673 | 0.7703 | 0.4869 | 0.6736 | 0.4790 | 0.5337 | **0.8888** | 0.7757 | 0.8466 |
|  | (0.033) | (0.032) | (0.001) | (0.030) | (0.023) | (0.034) | (0.028) | (0.029) | (0.030) | (0.126) | (0.009) | (0.029) | (0.016) | (0.005) | (0.017) | (0.046) | (0.029) |
| Spyware-Attacks (SPD) | 0.5082 | 0.4915 | 0.4914 | 0.4947 | 0.4855 | 0.4953 | 0.4611 | 0.4913 | **0.5273** | 0.4864 | 0.4838 | 0.4789 | 0.5077 | 0.4789 | 0.4949 | 0.4948 | 0.5184 |
|  | (0.018) | (0.018) | (0.011) | (0.016) | (0.011) | (0.013) | (0.023) | (0.032) | (0.020) | (0.006) | (0.027) | (0.009) | (0.023) | (0.019) | (0.007) | (0.016) | (0.016) |
| Bank | 0.5117 | 0.5145 | 0.5109 | 0.5125 | 0.5018 | 0.5123 | 0.5302 | 0.5127 | 0.5092 | 0.5242 | 0.5871 | 0.5107 | 0.5156 | 0.6463 | 0.5666 | 0.6169 | **0.6767** |
|  | (0.004) | (0.003) | (0.003) | (0.003) | (0.005) | (0.004) | (0.003) | (0.003) | (0.003) | (0.010) | (0.007) | (0.004) | (0.034) | (0.003) | (0.019) | (0.001) | (0.005) |
| Seismic | 0.6973 | 0.7018 | **0.7421** | 0.6970 | 0.6899 | 0.7289 | 0.7088 | 0.7171 | 0.6744 | 0.7151 | 0.7015 | 0.7245 | 0.7085 | 0.7393 | 0.6741 | 0.7003 | 0.7375 |
|  | (0.015) | (0.011) | (0.006) | (0.008) | (0.014) | (0.007) | (0.003) | (0.006) | (0.003) | (0.005) | (0.013) | (0.011) | (0.064) | (0.002) | (0.024) | (0.003) | (0.009) |
| CMC | 0.5198 | 0.5094 | 0.5677 | 0.5360 | 0.5701 | 0.5052 | 0.4959 | 0.4778 | 0.5403 | 0.5442 | 0.5401 | 0.5706 | 0.4671 | 0.5368 | 0.5746 | 0.5083 | **0.5789** |
|  | (0.010) | (0.004) | (0.015) | (0.007) | (0.034) | (0.006) | (0.009) | (0.009) | (0.012) | (0.018) | (0.038) | (0.046) | (0.093) | (0.020) | (0.016) | (0.004) | (0.009) |
| Damage-Report (DAMRE) | 0.6628 | 0.8790 | 0.9269 | 0.8391 | 0.9117 | 0.9133 | 0.7766 | 0.8850 | 0.8283 | 0.9221 | 0.8840 | 0.7780 | 0.6789 | **0.9393** | 0.8864 | 0.8968 | 0.9311 |
|  | (0.028) | (0.011) | (0.012) | (0.012) | (0.010) | (0.002) | (0.054) | (0.048) | (0.008) | (0.008) | (0.037) | (0.062) | (0.056) | (0.004) | (0.018) | (0.010) | (0.005) |
| OS-Kernel (OS) | 0.7825 | 0.8480 | 0.9271 | 0.7633 | 0.7991 | 0.8638 | 0.8650 | 0.9040 | 0.7966 | **0.9581** | 0.9386 | 0.9171 | 0.6735 | 0.9203 | 0.9408 | 0.8542 | 0.9471 |
|  | (0.022) | (0.014) | (0.009) | (0.012) | (0.025) | (0.007) | (0.013) | (0.017) | (0.027) | (0.015) | (0.011) | (0.019) | (0.076) | (0.009) | (0.012) | (0.012) | (0.006) |
| Smart-Meter (SMD) | 0.8132 | 0.9326 | 0.9238 | 0.9108 | 0.6750 | 0.9272 | 0.4793 | 0.1473 | 0.5166 | 0.8505 | 0.8057 | 0.8018 | 0.7222 | 0.9157 | 0.8393 | **0.9574** | 0.9369 |
|  | (0.009) | (0.003) | (0.002) | (0.002) | (0.039) | (0.003) | (0.029) | (0.012) | (0.022) | (0.015) | (0.020) | (0.021) | (0.065) | (0.003) | (0.045) | (0.004) | (0.008) |

## C.2. F1-score

We report F1-score as a complementary metric. Figure 9 summarizes the mean F1-score over all 28 datasets. Table 5 reports F1-score and standard deviation for each dataset. On average, our CausalAno is better than DRL around 6% and LLM-based methods around 7-8%.

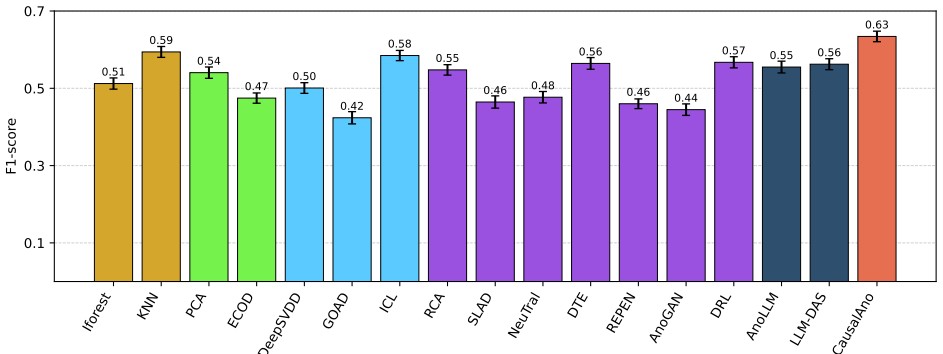

*Figure 9.* Average F1-scores for all methods over all 28 datasets (*higher is better*).

*Table 5.* F1-score (standard deviation) for each dataset (*higher is better*).

| Dataset | Iforest | PCA | KNN | ECOD | DeepSVDD | RCA | SLAD | GOAD | NeuTral | ICL | DTE | REPEN | AnoGAN | AnoLLM | DRL | LLM-DAS | CausalAno |
|---|---|---|---|---|---|---|---|---|---|---|---|---|---|---|---|---|---|
| Annthyroid | 0.5479 (0.019) | 0.4873 (0.029) | 0.4397 (0.005) | 0.3876 (0.003) | 0.4258 (0.022) | 0.3730 (0.005) | 0.4093 (0.007) | 0.2284 (0.003) | 0.4202 (0.012) | 0.5255 (0.033) | **0.7685** (0.020) | 0.3187 (0.026) | 0.3060 (0.087) | 0.6071 (0.007) | 0.5169 (0.011) | 0.4629 (0.024) | 0.6730 (0.011) |
| BreastW | **0.9652** (0.009) | 0.9585 (0.009) | 0.9635 (0.007) | 0.9469 (0.010) | 0.9295 (0.012) | 0.9552 (0.008) | 0.9527 (0.012) | 0.9635 (0.006) | 0.9619 (0.009) | 0.9594 (0.010) | 0.9560 (0.009) | 0.9278 (0.031) | 0.9278 (0.068) | 0.9635 (0.005) | 0.9569 (0.007) | 0.9527 (0.006) | 0.9552 (0.007) |
| Ecoli | 0.6445 (0.109) | **0.7778** (0.000) | 0.7334 (0.054) | 0.3777 (0.055) | **0.7778** (0.000) | 0.5778 (0.044) | 0.5556 (0.157) | 0.5111 (0.089) | 0.6445 (0.054) | 0.7334 (0.054) | **0.7778** (0.000) | 0.6000 (0.229) | 0.2444 (0.083) | **0.7778** (0.000) | 0.6445 (0.083) | 0.7111 (0.089) | 0.7334 (0.055) |
| Mammography | 0.3715 (0.028) | 0.4654 (0.011) | 0.4077 (0.018) | **0.5346** (0.004) | 0.4508 (0.052) | 0.3485 (0.006) | 0.1515 (0.022) | 0.2900 (0.010) | 0.1269 (0.009) | 0.2546 (0.013) | 0.3315 (0.023) | 0.2892 (0.017) | 0.3554 (0.109) | 0.3538 (0.010) | 0.3138 (0.021) | 0.5169 (0.010) | 0.3354 (0.020) |
| Shuttle | 0.9672 (0.005) | 0.9579 (0.001) | 0.9774 (0.001) | 0.9154 (0.002) | 0.9599 (0.038) | 0.9696 (0.001) | 0.9755 (0.001) | 0.9618 (0.001) | 0.9830 (0.002) | 0.9794 (0.002) | 0.9740 (0.002) | 0.9675 (0.002) | 0.9540 (0.003) | 0.9810 (0.000) | 0.9820 (0.001) | 0.9730 (0.001) | **0.9876** (0.001) |
| Thyroid | 0.7892 (0.009) | 0.7355 (0.023) | 0.6366 (0.008) | 0.6344 (0.014) | 0.5204 (0.062) | 0.6129 (0.015) | 0.6495 (0.045) | 0.4280 (0.011) | 0.2688 (0.045) | 0.7269 (0.031) | **0.8021** (0.013) | 0.3871 (0.034) | 0.5828 (0.067) | 0.7978 (0.013) | 0.7226 (0.057) | 0.6215 (0.028) | 0.7419 (0.035) |
| Wine | 0.9381 (0.007) | 0.9803 (0.007) | 0.9746 (0.006) | 0.8113 (0.019) | 0.9408 (0.027) | 0.9437 (0.013) | **0.9859** (0.015) | 0.9803 (0.011) | 0.9746 (0.006) | 0.9577 (0.000) | 0.9774 (0.007) | 0.6085 (0.033) | 0.9521 (0.021) | 0.9803 (0.007) | 0.9803 (0.007) | 0.9690 (0.011) | 0.9831 (0.006) |
| Waveform | 0.0960 (0.022) | 0.0860 (0.005) | 0.2940 (0.010) | 0.0760 (0.005) | 0.0800 (0.025) | 0.1400 (0.006) | 0.0320 (0.010) | 0.0340 (0.014) | **0.4540** (0.024) | 0.3500 (0.078) | 0.1140 (0.034) | 0.2680 (0.069) | 0.0640 (0.023) | 0.0900 (0.011) | 0.1940 (0.060) | 0.1080 (0.021) | 0.4460 (0.045) |
| WBC | **0.8800** (0.040) | 0.8400 (0.050) | **0.8800** (0.040) | 0.8400 (0.050) | 0.4400 (0.102) | 0.8000 (0.063) | 0.8200 (0.075) | 0.7400 (0.049) | 0.3000 (0.141) | 0.7000 (0.110) | 0.4800 (0.172) | 0.7200 (0.040) | **0.8800** (0.040) | 0.8600 (0.050) | 0.6600 (0.136) | 0.8200 (0.040) | 0.7400 (0.080) |
| Pendigits | 0.5897 (0.031) | 0.4436 (0.010) | **0.9192** (0.013) | 0.4282 (0.006) | 0.4025 (0.082) | 0.5667 (0.031) | 0.3474 (0.073) | 0.0000 (0.000) | 0.4513 (0.106) | 0.6320 (0.074) | 0.5859 (0.074) | 0.3667 (0.175) | 0.4487 (0.013) | 0.4782 (0.075) | 0.6436 (0.074) | 0.4936 (0.074) | 0.8654 (0.019) |
| Cardio | 0.7216 (0.021) | **0.8057** (0.009) | 0.6591 (0.013) | 0.6648 (0.017) | 0.5818 (0.037) | 0.7511 (0.021) | 0.6011 (0.012) | 0.2591 (0.040) | 0.5352 (0.016) | 0.6477 (0.043) | 0.6580 (0.039) | 0.5852 (0.030) | 0.6364 (0.094) | 0.6171 (0.014) | 0.6773 (0.044) | 0.7648 (0.025) | 0.7091 (0.031) |
| PageBlocks | 0.6502 (0.005) | 0.6427 (0.004) | 0.7529 (0.012) | 0.5702 (0.007) | 0.6765 (0.009) | 0.6718 (0.007) | 0.5647 (0.038) | 0.6431 (0.013) | 0.6541 (0.021) | 0.7314 (0.011) | 0.7624 (0.017) | 0.6337 (0.009) | 0.4745 (0.083) | 0.5820 (0.007) | 0.7965 (0.016) | 0.6392 (0.007) | **0.8267** (0.003) |
| Stamps | 0.6903 (0.048) | 0.6258 (0.056) | 0.6129 (0.058) | 0.5097 (0.094) | 0.5032 (0.097) | 0.6000 (0.044) | 0.2452 (0.056) | 0.1613 (0.035) | 0.4645 (0.066) | 0.5355 (0.105) | 0.5355 (0.066) | 0.4516 (0.127) | 0.4581 (0.231) | 0.6581 (0.033) | 0.6322 (0.044) | 0.5871 (0.077) | **0.6968** (0.063) |
| Glass | 0.1111 (0.000) | 0.1111 (0.000) | 0.1778 (0.054) | 0.1333 (0.044) | 0.2444 (0.083) | 0.1778 (0.089) | 0.1111 (0.000) | 0.1333 (0.044) | 0.3333 (0.141) | **0.4000** (0.166) | 0.1555 (0.054) | 0.0667 (0.089) | 0.1333 (0.083) | 0.1111 (0.000) | 0.2000 (0.044) | 0.1111 (0.000) | 0.2444 (0.109) |
| Vowels | 0.2960 (0.023) | 0.1920 (0.010) | 0.6840 (0.039) | 0.2200 (0.000) | 0.5560 (0.060) | 0.4240 (0.020) | 0.7120 (0.020) | 0.4560 (0.051) | **0.8120** (0.016) | 0.7400 (0.042) | 0.7840 (0.045) | 0.2640 (0.015) | 0.0880 (0.039) | 0.2320 (0.024) | 0.8080 (0.050) | 0.2520 (0.016) | 0.8320 (0.043) |
| MagicGamma | 0.7014 (0.005) | 0.6541 (0.001) | 0.7705 (0.002) | 0.6030 (0.002) | 0.6625 (0.008) | 0.6923 (0.002) | 0.6769 (0.003) | 0.6076 (0.004) | 0.7506 (0.007) | 0.7162 (0.011) | **0.7809** (0.015) | 0.7048 (0.016) | 0.6357 (0.028) | 0.6547 (0.002) | 0.7341 (0.008) | 0.6554 (0.001) | 0.7647 (0.012) |
| Yeast | **0.4821** (0.030) | 0.4463 (0.020) | 0.3453 (0.012) | 0.3368 (0.000) | 0.2484 (0.032) | 0.3705 (0.012) | 0.0863 (0.031) | 0.0105 (0.000) | 0.0863 (0.043) | 0.2779 (0.022) | 0.2695 (0.058) | 0.2274 (0.066) | 0.3768 (0.200) | 0.3158 (0.015) | 0.3032 (0.039) | 0.3663 (0.034) | 0.3789 (0.023) |
| ImgSeg | 0.7432 (0.014) | 0.7236 (0.004) | 0.8537 (0.005) | 0.6610 (0.004) | 0.8517 (0.007) | 0.7513 (0.007) | 0.8287 (0.011) | 0.7709 (0.003) | 0.8606 (0.002) | 0.8586 (0.028) | 0.8232 (0.011) | 0.6418 (0.011) | 0.6794 (0.043) | 0.7051 (0.006) | 0.8412 (0.014) | 0.7396 (0.004) | **0.8731** (0.005) |
| Lymphography | 0.2000 (0.066) | 0.6334 (0.067) | 0.6667 (0.000) | 0.3000 (0.125) | 0.6000 (0.082) | 0.7333 (0.082) | 0.7000 (0.067) | 0.6000 (0.082) | 0.6000 (0.133) | 0.7000 (0.067) | 0.7333 (0.082) | 0.6334 (0.067) | 0.7667 (0.082) | **0.9000** (0.082) | 0.8000 (0.125) | **0.9000** (0.082) | 0.7000 (0.067) |
| Cybersecurity (ACD) | 0.0869 (0.006) | 0.0869 (0.009) | 0.0878 (0.007) | 0.1008 (0.012) | 0.0808 (0.012) | 0.0930 (0.011) | 0.0906 (0.005) | 0.0918 (0.008) | 0.0975 (0.019) | 0.1098 (0.004) | 0.1086 (0.012) | 0.0869 (0.010) | 0.0902 (0.007) | **0.1143** (0.007) | 0.0910 (0.011) | 0.1025 (0.009) | 0.1094 (0.012) |
| NHIS-Claims | 0.1786 (0.040) | 0.1178 (0.040) | 0.2714 (0.021) | 0.1393 (0.018) | 0.2500 (0.028) | 0.2893 (0.021) | 0.0893 (0.034) | 0.2821 (0.007) | 0.0964 (0.024) | **0.3036** (0.020) | 0.0179 (0.016) | 0.2750 (0.009) | 0.0179 (0.000) | 0.0000 (0.000) | 0.1250 (0.025) | 0.2715 (0.018) | 0.2072 (0.035) |
| Spyware-Attacks (SPD) | 0.8678 (0.004) | 0.8649 (0.004) | 0.8670 (0.003) | 0.8660 (0.004) | 0.8621 (0.002) | 0.8657 (0.004) | 0.8626 (0.004) | 0.8670 (0.004) | **0.8709** (0.004) | 0.8657 (0.004) | 0.8657 (0.004) | 0.8623 (0.003) | 0.8654 (0.004) | 0.8647 (0.001) | 0.8662 (0.004) | 0.8605 (0.003) | 0.8644 (0.002) |
| Bank | 0.2134 (0.004) | 0.2198 (0.003) | 0.2146 (0.003) | 0.2183 (0.003) | 0.2057 (0.007) | 0.2130 (0.006) | 0.2269 (0.004) | 0.2164 (0.004) | 0.2094 (0.002) | 0.2337 (0.014) | 0.3250 (0.006) | 0.2198 (0.004) | 0.2218 (0.034) | 0.3957 (0.002) | 0.2642 (0.019) | 0.3536 (0.001) | **0.4247** (0.006) |
| Seismic | 0.2576 (0.028) | 0.2871 (0.014) | 0.3000 (0.032) | 0.2765 (0.008) | 0.2259 (0.016) | 0.2941 (0.015) | 0.2859 (0.025) | 0.2894 (0.009) | 0.2059 (0.026) | 0.2882 (0.020) | 0.2317 (0.021) | 0.2988 (0.010) | 0.3047 (0.023) | **0.3212** (0.017) | 0.2306 (0.022) | 0.2894 (0.017) | 0.3024 (0.021) |
| CMC | 0.1310 (0.026) | 0.0759 (0.014) | 0.1034 (0.022) | 0.0690 (0.000) | 0.0897 (0.041) | 0.1172 (0.017) | 0.1241 (0.017) | 0.0965 (0.014) | **0.1586** (0.028) | 0.0896 (0.017) | 0.1034 (0.022) | 0.0897 (0.028) | 0.0621 (0.055) | 0.1103 (0.026) | 0.0896 (0.035) | 0.0690 (0.000) | 0.0965 (0.034) |
| Damage-Report (DAMRE) | 0.3680 (0.041) | 0.7140 (0.037) | 0.8280 (0.018) | 0.6520 (0.012) | 0.7560 (0.058) | 0.8220 (0.008) | 0.3860 (0.122) | 0.6360 (0.023) | 0.5200 (0.035) | 0.7860 (0.021) | 0.7160 (0.085) | 0.4560 (0.088) | 0.3300 (0.067) | **0.8680** (0.004) | 0.6860 (0.027) | 0.8220 (0.015) | 0.8420 (0.010) |
| OS-Kernel (OS) | 0.5055 (0.023) | 0.6374 (0.017) | 0.6857 (0.030) | 0.5121 (0.017) | 0.5121 (0.044) | 0.6527 (0.024) | 0.4923 (0.011) | 0.6066 (0.039) | 0.4132 (0.057) | 0.8110 (0.066) | 0.7802 (0.025) | 0.6198 (0.037) | 0.3253 (0.089) | 0.6879 (0.019) | 0.7297 (0.052) | 0.6374 (0.012) | **0.8637** (0.005) |
| Smart-Meter (SMD) | 0.3504 (0.016) | 0.5624 (0.017) | 0.5280 (0.010) | 0.5056 (0.014) | 0.1864 (0.030) | 0.5272 (0.024) | 0.0440 (0.009) | 0.0000 (0.000) | 0.1000 (0.016) | 0.4600 (0.031) | 0.3848 (0.033) | 0.3096 (0.024) | 0.2720 (0.051) | 0.5120 (0.018) | 0.3936 (0.074) | **0.6960** (0.030) | 0.5608 (0.020) |

## C.3. Rank

We further summarize each method's performance using an **average-rank** protocol based on the AUC-ROC scores. For each dataset, we rank all methods from **1 (best)** to **17 (worst)** according to AUC-ROC, then show average ranks across datasets in Figure 10.

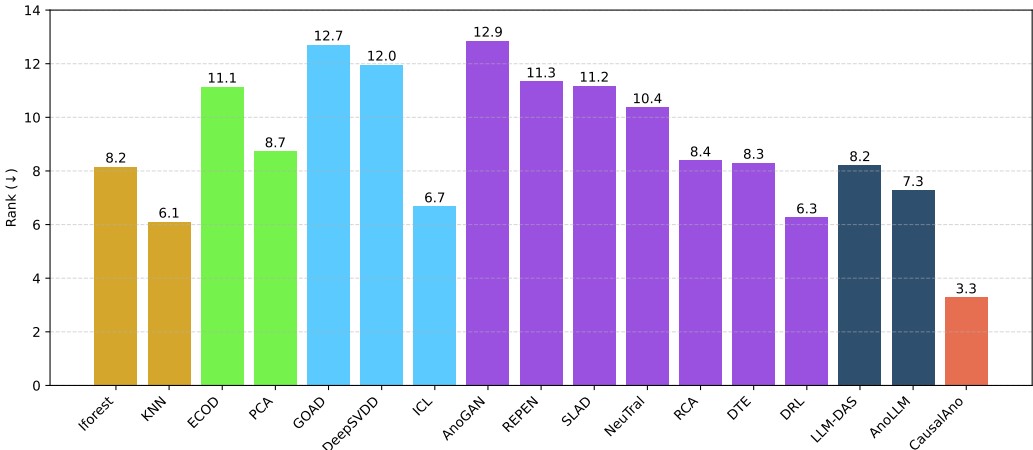

*Figure 10.* Average ranks across all 28 datasets (*lower is better*).

# D. Additional Ablation Studies

We further analyze the robustness of our CausalAno with respect to two training hyper-parameters: *batch size* and the *causal-alignment weight* $\lambda$. Figure 11 and Figure 12 report AUC-ROC trends under different configurations.

## D.1. Batch size

As shown in Figure 11, CausalAno's performance remains stable with a large range of batch sizes in $[64, 512]$. More importantly, it always outperforms DRL.

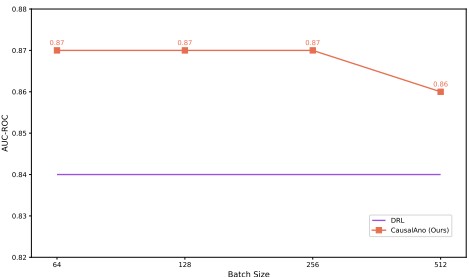

*Figure 11.* AUC-ROC of CausalAno with different batch sizes.

## D.2. Causal-alignment weight $\lambda$

As shown in Figure 12, $\lambda$ critically controls the balance between the adversarial objective and the causal-alignment objective, and thus has a clear impact on our AUC-ROC.

When $\lambda = 0$, CausalAno is trained purely with the adversarial loss and receives no causal feedback, meaning it cannot capture the causal knowledge. This setting yields the worst performance at 0.81 AUC-ROC score. CausalAno is stable and effective when $\lambda \in [0.01, 0.1]$, where causal guidance complements adversarial learning without overwhelming it. When $\lambda$ is large e.g., $\lambda > 0.1$, the causal loss may become numerically large since SHD metric is proportional to the number of features $M \times M$. As a result, it may dominate the adversarial loss, weakening the adversarial learning signal and degrading performance. However, with a small enough $\lambda$, CausalAno is always better than the best baseline DRL.

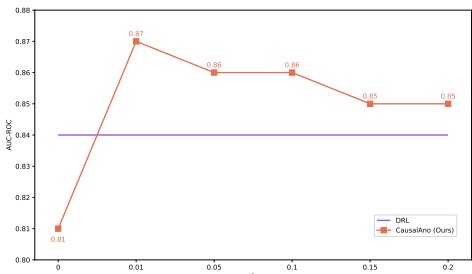

*Figure 12.* AUC-ROC of CausalAno with different $\lambda$ values.

# E. Computational Efficiency

We report the average wall-clock runtime (in *minutes*) of each method over 28 datasets in Figure 13. All runtime results are measured on a computer with *GPU: NVIDIA GeForce RTX 4070 Super, CPU: 20 cores, Memory: 16GB*.

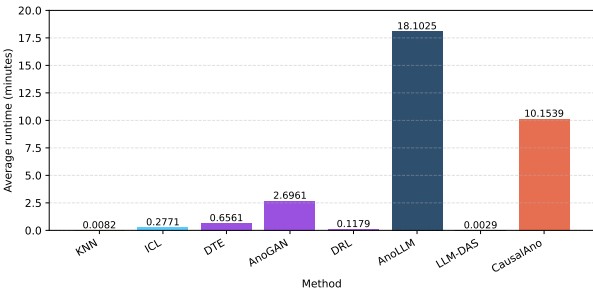

*Figure 13.* Average runtime (in *minutes*) of top methods over all 28 datasets.

# F. Robustness to Causal Graph

We evaluate how sensitive CausalAno is to the correctness of the estimated causal graph $\mathcal{G}_{real}$ obtained by running PC on normal samples $\mathcal{D}_{real}$. We systematically perturb the ground-truth causal graph to create a graph-quality spectrum as shown in Table 6. For each graph-accuracy level in $\{0\%, 20\%, 40\%, 80\%, 100\%\}$, we report the number of anomalies found by CausalAno i.e., *true positives* (TP).

*Table 6.* Graph-quality spectrum for a three-variable causal graph derived from the Synthetic dataset (described in Section 4.3.3 of the main paper). A, E, and S denote Age, Edu, and Salary.

| Graph-accuracy level | Graph structure | Note |
|---|---|---|
| | | Ground truth |
| 100% | | The true mechanism |
| 80% | | Missed one parent |
| 60% | | No edges found |
| 40% | | One wrong chain |
| 20% | | Effect treated as cause |
| 0% | | Predicting causes from effect |

As shown in Figure 14, CausalAno is best performing when $\mathcal{G}_{real}$ is close to ground-truth causal structure, and its detection performance degrades as the graph-accuracy decreases. In the worst case (graph-accuracy is 0%), CausalAno's TP drops from 813 to 788, indicating that severe graph misspecification can harm our detection performance. However, even under this extreme setting, CausalAno remains clearly better than the DRL baseline (its TP is 710), confirming that our method maintains a strong advantage despite imperfect causal graphs.

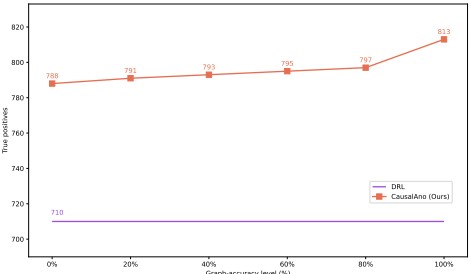

*Figure 14.* CausalAno's true positives vs. $\mathcal{G}_{real}$'s quality. The graph-accuracy levels are described in Table 6.

## G. Statistics of 18 Continuous Tabular Datasets

We summarize the characteristics of 18 continuous tabular datasets in Table 7.

*Table 7.* Statistics for 18 continuous datasets. $M_{cat}$, $M_{con}$, and $N_a$ are the numbers of categorical, continuous features, and anomalies.

| Dataset | $N$ | $M$ | $M_{cat}$ | $M_{con}$ | $N_a$ |
|---|---|---|---|---|---|
| Annthyroid | 7,200 | 6 | 0 | 6 | 534 (7%) |
| BreastW | 699 | 9 | 0 | 9 | 241 (34%) |
| Ecoli | 336 | 7 | 0 | 7 | 9 (3%) |
| Mammography | 11,183 | 6 | 0 | 6 | 260 (2%) |
| Shuttle | 49,097 | 9 | 0 | 9 | 3,511 (7%) |
| Thyroid | 3,772 | 6 | 0 | 6 | 93 (2%) |
| Wine | 178 | 13 | 0 | 13 | 71 (40%) |
| Waveform | 3,443 | 21 | 0 | 21 | 100 (3%) |
| WBC | 223 | 9 | 0 | 9 | 10 (4%) |
| Pendigits | 6,870 | 16 | 0 | 16 | 156 (2%) |
| Cardio | 1,831 | 21 | 0 | 21 | 176 (10%) |
| PageBlocks | 5,393 | 10 | 0 | 10 | 510 (9%) |
| Stamps | 340 | 9 | 0 | 9 | 31 (9%) |
| Glass | 214 | 7 | 0 | 7 | 9 (4%) |
| Vowels | 1,456 | 12 | 0 | 12 | 50 (3%) |
| MagicGamma | 19,020 | 10 | 0 | 10 | 6,688 (35%) |
| Yeast | 1,484 | 8 | 0 | 8 | 95 (6%) |
| ImgSeg | 2,310 | 18 | 0 | 18 | 990 (43%) |

