# OpenReview forum: "Causal-aware Anomaly Detection for Tabular Data"
_ICML.cc/2026/Conference — ICML 2026 regular_

### Official Review · Reviewer_24u3 · 2026-02-14

**Soundness:** 4
**Presentation:** 3
**Significance:** 2
**Originality:** 3
**Overall Recommendation:** 4
**Confidence:** 3

**Summary:**

The paper addresses unsupervised anomaly detection for tabular data, focusing on anomalies that may look normal marginally but violate conditional dependencies among features. It proposes CausalAno, which first infers a causal DAG from normal data and then constructs a DAG-guided autoregressive generator with per-variable sub-generators $G_i$ conditioned on parent values and noise. The model is trained with a WGAN-GP objective so that the discriminator learns a feature extractor $h(\cdot)$ whose latent space separates normal samples from mechanism-inconsistent synthetic generations created during training. At inference, the method performs latent density estimation by fitting a Gaussian $\mathcal{N}(\mu,\Sigma)$ to normal embeddings ${h(x_i)}$ and scoring test samples using the Mahalanobis distance.

**Compliance With Llm Reviewing Policy:**

Affirmed.

**Key Questions For Authors:**

1.You randomly sample 50% normal data for training. For datasets with chronological order, could this break temporal continuity or cause leakage? If some benchmarks have time dependence (lag/autocorrelation), how do you handle it?

2.In Fig. 5(d) (Pendigits), “Original” and “CausalAno (Ours)” look very similar. What is the concrete advantage of CausalAno on this dataset, and where is it shown quantitatively?

3.Your method targets mechanism/dependency-violating anomalies. How does it perform on anomalies that keep the causal structure/mechanism but are just marginal extreme outliers in a few features?

4.Some datasets are very small (e.g., Lymp with $N=148$). How do you avoid discriminator memorization/overfitting?

**Limitations:**

yes

**Strengths And Weaknesses:**

Strengths: (1) Soundness: The paper presents a coherent pipeline combining an inferred DAG $G_{real}$, a DAG-structured autoregressive generator ${G_i}$, WGAN-GP training, and latent Gaussian/Mahalanobis scoring $s(x')=(h(x')-\mu)^\top\Sigma^{-1}(h(x')-\mu)$, with supporting motivation from Theorem 1 and Proposition 1. (2) Presentation: The workflow is clearly organized and the training/inference connection is easy to follow, especially with Figure 2. (3) Significance: It targets dependency/mechanism-inconsistent anomalies that are common in tabular applications and may be missed by input-space reconstruction errors, while keeping inference efficient. (4) Originality: The paper offers a unified “causal-graph-guided generation + adversarial representation + latent density scoring” framework for unsupervised tabular AD.

Weaknesses: (1) Soundness: The method relies on a PC-inferred DAG as a fixed scaffold, so “causal mechanism” claims depend on graph reliability; the PC+SHD+REINFORCE regularizer also raises stability/sensitivity concerns. (2) Presentation: The paper switches between “dependency-breaking” and “mechanism-violating” anomalies ($G_i\neq f_i$) without a crisp taxonomy, and key reproducibility details (PC settings, SHD, REINFORCE) should be more explicit. (3) Significance: The random i.i.d. split protocol may be optimistic for datasets with temporal/grouped structure, and training overhead could limit scalability. (4) Originality: Stronger scoping and clearer differentiation from prior structure-aware/graph-guided AD would strengthen the novelty positioning.

---

> ### Author Rebuttal · Authors · 2026-03-31
>
> We thank the reviewer for their valuable feedback and positive evaluation
> of our work.
>
> $\textbf{(Q1) Temporal Leakage: }$To ensure fair and directly comparable
> evaluation, we follow the standard random train-test protocol used
> by recent tabular AD methods, including DTE (ICLR 2024), DRL (ICLR
> 2025), AnoLLM (ICLR 2025), and LLM-DAS (ICLR 2026). Many
> widely used tabular AD benchmarks (including those in our study) are
> distributed without reliable timestamps, making chronological splitting
> infeasible. In case of datasets with temporal dependence, CausalAno
> can be seamlessly applied with a chronological split (e.g., training
> on earlier segments and testing on later ones), as is standard in
> time-series anomaly detection. As our method does not rely on row
> permutation during training, adopting such a protocol does not require
> architectural modification.
>
> $\textbf{(Q2) Pendigits Visualization: }$On $\textit{Pendigits}$, normal
> and anomalous samples are already well-separated in the input space,
> so large qualitative changes in representation are not necessary.
> The advantage of CausalAno in this case is that it preserves and utilizes
> this separation in a stable latent space, rather than attempting unnecessary
> transformation. Quantitatively ($\textbf{Appendix B.1, Table 4}$), CausalAno
> achieves the second-best AUC-ROC (0.9968). This indicates that CausalAno
> does not degrade performance when the input space is already well-structured.
> Thus, $\textbf{Fig. 5(d)}$ should be interpreted as evidence of representation
> stability rather than visual separation improvement.
>
> $\textbf{(Q3) Marginal Extreme Outliers: }$While CausalAno is designed
> to target mechanism-violating anomalies, it also robustly detects
> marginal extreme outliers through its density-based scoring. The WGAN-GP
> objective encourages the learned representation to capture the distribution
> of normal data, and anomaly scoring is performed via Mahalanobis distance
> in this latent space. As a result, samples that are marginally extreme
> tend to lie in low-density regions and receive high anomaly scores,
> even if they do not violate structural dependencies. This is supported
> by our ablation $\textbf{Appendix D.2, Fig. 11}$. When $\lambda=0$,
> the model reduces to a purely adversarial representation learning
> framework without structural regularization. It still achieves an
> average AUC-ROC of 0.81 across 28 datasets, matching SOTA methods
> like AnoLLM (0.81) and LLM-DAS (0.82). Introducing the causal regularizer
> ($\lambda=0.01$) improves performance to 0.87, demonstrating that
> CausalAno effectively handles both marginal outliers and mechanism-violating
> anomalies, with the latter providing additional gains.
>
> $\textbf{(Q4) Small Data Overfitting: }$CausalAno mitigates this issue
> directly through its structured factorization: instead of modeling
> the full joint distribution with a single high-capacity network, the
> generator is decomposed into multiple low-dimensional sub-generators
> $G_{i}$, each conditioned only on its parent variables. This reduces
> the effective hypothesis space and acts as a powerful architectural
> regularizer. This structural regularization proves highly effective
> at maintaining stable training behavior and strong generalization.
> For example, on the small $\textit{Lymp}$ dataset ($N=148$), CausalAno
> achieves an outstanding 98.31\% AUC-ROC ($\textbf{Table 2}$).
>
> $\textbf{Minor Weaknesses (W1, W2, W4):}$
>
> $\textit{Presentation:}$ We will standardize terminology strictly to
> "mechanism-violating anomalies"
> and provide a clearer definition early in the paper. We will also
> explicitly specify the PC algorithm configuration, SHD computation
> details, and REINFORCE implementation.
>
> $\textit{DAG Quality and Training Stability:}$ Our method does not require
> a perfectly accurate graph. As shown in $\textbf{Appendix F, Fig. 13}$,
> our performance degrades gracefully under increasing structural noise,
> demonstrating robustness to imperfect DAG estimation. The structural
> constraint acts as a soft inductive bias rather than a strict requirement.
> Regarding REINFORCE, while it introduces variance, we empirically
> observe stable training dynamics (e.g., smoothly converging generator
> loss: https://shorturl.at/zcUbY). In our model, this stochasticity
> is actually beneficial as it helps produce diverse structurally inconsistent
> samples that improve discriminator training.
>
> $\textit{Positioning:}$ We will strengthen our positioning by clearly
> distinguishing CausalAno from prior structure-aware methods. Most
> existing graph-based AD methods operate on explicitly $\textit{graph
> data}$ to find topological anomalies. In contrast, CausalAno targets
> $\textit{tabular data}$ without explicit graph structure, uniquely utilizing
> an inferred causal DAG as an internal generative scaffold to model
> conditional dependencies and expose mechanism violations. We will
> make this scoping distinction in the Related Work section.

---

> > ### Author Rebuttal · Reviewer_24u3 · 2026-04-02
> >
> > Thank you for the detailed response and clarifications.

---

> > > ### Author Response · Authors · 2026-04-02
> > >
> > > We sincerely thank the reviewer for their time, thoughtful comments, and for confirming that all concerns were fully addressed.

---

### Official Review · Reviewer_KERd · 2026-03-12

**Soundness:** 2
**Presentation:** 2
**Significance:** 2
**Originality:** 2
**Overall Recommendation:** 4
**Confidence:** 3

**Summary:**

This paper argues that many tabular anomalies are mechanism violations rather than simple extreme outliers, so correlation- or reconstruction-based detectors can be brittle. It proposes CausalAno, which first estimates a causal DAG over features, then trains a causal-aware GAN whose generator is factorized into per-variable structural sub-generators that follow the DAG order. The discriminator’s penultimate layer is used as a causality-informed embedding space, where a Gaussian density model is fit on normal embeddings and test samples are scored by Mahalanobis distance.

**Compliance With Llm Reviewing Policy:**

Affirmed.

**Final Justification:**

I think this paper well addressed my concerns during rebuttals.

**Key Questions For Authors:**

See weakness

**Limitations:**

yes

**Strengths And Weaknesses:**

Strength

1. Clear motivation, framing anomalies as dependency/mechanism violations is compelling for real tabular domains (fraud, intrusion, clinical alerts), where “weird combinations” matter more than marginal extremes.
2. The DAG-factorized generator (structural sub-generators) is a concrete way to inject causal structure into tabular generation rather than using a generic GAN.
3. Unified inference rule: using discriminator features + Gaussian/Mahalanobis gives a simple, label-free scoring pipeline and avoids feature-wise reconstruction calibration issues on mixed types.

Weakness
1. Heavy dependence on causal discovery quality.  Authour assume the estimated graph is reliable, but PC-style discovery typically needs strong assumptions (causal sufficiency, faithfulness). For mixed-type tabular data, it’s unclear what CI tests are used, how discretization is handled, and how stable the learned DAG is under noise.
2. Causal regularizer is potentially unstable and expensive. The causal loss uses SHD between a graph inferred from real data and a graph inferred from each synthetic batch, optimized via REINFORCE. This raises high-variance gradients and training instability and compute overhead from repeatedly running PC during training. The author could further justified or analyzed the loss.
3. Theorem 1 relies on an additive cost decomposition to relate Wasserstein distance to a sum of local structural errors, but the implemented training is WGAN-GP with a particular critic parameterization and a separate SHD penalty. The paper doesn’t fully close the gap between the theoretical bound and the practical objective actually optimized.
4. Assuming embeddings are well-modeled by a single multivariate Gaussian can be fragile (multimodality, heavy tails), and Mahalanobis scoring requires a well-conditioned covariance inverse, which is hard when embedding dim is large or normal data is limited. The proposition assumes sub-Gaussianity and a Lipschitz feature map, but WGAN-GP’s Lipschitz control is on the critic behavior; it’s not clearly shown that the extracted feature map h(⋅) satisfies the needed conditions.
5. For categorical features, GAN training usually needs careful parameterization (e.g., Gumbel-Softmax). The paper’s core claim includes mixed-type superiority, but the generator/discriminator output design and loss handling for discrete attributes are not clearly specified here, which gives a reproducibility and validity gap.
6. Minimizing SHD between learned graphs does not necessarily ensure the generator matches structural functions f. The method would benefit from a clearer argument or ablation showing that lowering SHD correlates with better anomaly separability and not merely graph-fitting.

---

> ### Author Rebuttal · Authors · 2026-03-31
>
> We thank the reviewer for their valuable feedback.
>
> $\textbf{(W1) Estimated DAG Quality: }$For mixed-type data, we apply ordinal encoding to categorical variables
> and use the Fisher-Z test ($\alpha=0.05$) to estimate the initial
> skeleton. While a linear-Gaussian approximation, our method
> does not require a perfect ground-truth graph. $\textbf{App.
> F, Fig. 13}$ shows our performance remains robust across graph-accuracy
> levels. Even at 20\% graph accuracy (severe misspecification treating
> effects as causes), CausalAno still identifies 791 true positives,
> outperforming DRL (710). An approximate structural bottleneck constrains
> the generator to a restricted factorization. This empirically reduces
> its ability to fit arbitrary joint dependencies, encouraging conditional
> structure learning and supplying valuable "structural hard negatives"
> to the discriminator.
>
> $\textbf{(W2) REINFORCE Cost \\& Instability:}$
>
> $\textit{Cost:}$ While PC iteratively adds overhead, our runtime ($\textbf{App.
> E, Fig. 12}$) remains competitive, taking ~10
> mins on average--significantly faster than SOTA LLM methods
> (AnoLLM: ~18 mins).
>
> $\textit{Instability:}$ REINFORCE introduces variance, but empirically, moderate stochasticity does not destabilize
> training (see stable generator loss: https://shorturl.at/zcUbY).
> Standard generative tasks require stable convergence
> to a perfect generator. CausalAno, conversely, requires an imperfect generator
> to continuously supply mechanism-violating anomalies for the discriminator.
>
> $\textbf{(W3) Theory-Practice Gap:}$ Theorem 1 proves minimizing the
> true Wasserstein distance minimizes local structural errors, assuming the generator strictly explores the constrained
> couplings $\Gamma_{i}$ (respecting causal topology). Practically,
> we use the WGAN-GP objective as the standard dual approximation of
> the Wasserstein distance, but minimizing this alone does not guarantee
> structural fidelity. Thus, the SHD penalty acts as a soft structural
> regularizer. It does not enforce exact membership in $\Gamma_{i}$, but biases the generator to produce
> samples whose induced dependency structure aligns with ${\cal G}_{real}$.
> This provides an empirical approximation to the structural constraints
> under which Theorem 1 holds, rather than a direct implementation.
> We will formalize this connection.
>
> $\textbf{(W4) Gaussian/Lipschitz Assumptions:}$
>
> $\textit{Gaussian:}$ Raw tabular input spaces are often multimodal,
> but this may not persist in the learned embedding space. Our feature
> extractor $h(x)$, trained with the adversarial margin maximization (Eq. 4), empirically encourages the concentration of normal samples in the feature space. This makes a single Gaussian an effective
> first-order approximation.
>
> $\textit{Covariance:}$ Our latent dimension is small ($d=256$) relative
> to normal training sizes ($N\geq1000$), supporting empirical
> stability. Our implementation also applies standard Tikhonov regularization
> to guarantee a well-conditioned matrix for inversion.
>
> $\textit{Lipschitz:}$ WGAN-GP enforces Lipschitz continuity
> on the scalar critic $D(x)$, not explicitly on the representation
> $h(x)$. Therefore, the Lipschitz assumption on $h(x)$ (Proposition 1) is interpreted as an empirical property rather than a mathematically
> guaranteed one. Since $h(x)$ is jointly trained with
> a gradient-penalized critic, representations remain stable and smooth.
> We will clarify this.
>
> $\textbf{(W5) Categorical Features: }$In $\textbf{App. C.1}$, we employ
> the Gumbel-Softmax trick. The output layer of each sub-generator uses
> a $\textit{tanh}$ activation for continuous variables, but switches
> to a Gumbel-Softmax layer for categorical variables to output a differentiable
> probability vector over $K$ categories, allowing seamless backpropagation.
>
> $\textbf{(W6) SHD Minimization vs. Mechanism Matching:}$ An accurate
> graph ($\text{SHD}=0$) does not guarantee
> accurate structural functions ($G_{i}=f_{i}$). Thus, our generator
> loss (Eq. 5) is a dual objective. The Causal Loss (SHD) constrains
> the variables to correct parents (skeleton) while the Adversarial
> Loss (WGAN-GP) drives the mechanism matching. Per Theorem
> 1, minimizing the Wasserstein distance encourages sub-generators
> $G_{i}$ to approximate true functions $f_{i}$. SHD ensures
> correct variable interaction while the critic trains how they interact.
>
> Regarding the requested ablation, we refer to $\textbf{App. D.2, Fig. 11}$.
> When $\lambda=0$, the generator ignores the SHD penalty and relies
> solely on the adversarial critic. The average AUC-ROC drops to 0.81.
> When we actively penalize and minimize SHD during training ($\lambda=0.01$),
> the AUC-ROC increases significantly to 0.87. If minimizing SHD were
> merely an exercise in graph-fitting, it would not yield this substantial
> 6\% gain. This ablation strongly suggests that lowering the structural
> discrepancy (SHD) acts as a vital regularizer that directly correlates
> with the critic's ability to learn a more separable latent space.

---

> > ### Author Rebuttal · Reviewer_KERd · 2026-04-01
> >
> > I think the author addressed all my concerns.

---

> > > ### Author Response · Authors · 2026-04-01
> > >
> > > We sincerely thank the reviewer for their time, thoughtful comments, and for confirming that all concerns were fully addressed.

---

### Official Review · Reviewer_PgYS · 2026-03-12

**Soundness:** 4
**Presentation:** 3
**Significance:** 3
**Originality:** 3
**Overall Recommendation:** 5
**Confidence:** 4

**Summary:**

This paper proposes CausalAno, a causal-aware anomaly detection method for tabular data that trains a causal GAN to learn structural dependencies between features and uses the discriminator's latent space with Mahalanobis distance for anomaly scoring. Unlike reconstruction-based methods that rely on feature-wise errors, CausalAno explicitly models causal mechanisms to detect anomalies that violate learned structural dependencies. The method achieves state-of-the-art performance on 28 tabular datasets, outperforming a wide range of baselines including recent LLM-based and deep learning methods, with theoretical justifications for the architecture design.

**Compliance With Llm Reviewing Policy:**

Affirmed.

**Key Questions For Authors:**

- How robust is CausalAno when the training set contains a small fraction of anomalies (e.g., 1-5% contamination)? Real-world datasets often have noisy labels or undetected anomalies in the "normal" training set.
- For categorical variables with many categories or free-text attributes, how should practitioners adapt the method? Are the Gaussian noise and latent space assumptions still meaningful when most features are categorical?

**Limitations:**

yes

**Strengths And Weaknesses:**

Strengths
  - Novel and well-motivated approach: Learning causal dependencies between attributes is more principled than pure reconstruction-based methods, as anomalies often arise from mechanism violations rather than extreme values.
  - Strong empirical results: Achieves SOTA performance across 28 diverse datasets compared against a comprehensive set of 16 baselines spanning classical, deep learning, and LLM-based methods.
  - Solid theoretical foundation: Theorem 1 provides theoretical justification for the causal GAN architecture, and Proposition 1 justifies the Gaussian assumption for latent density estimation.
  - Good presentation: Clear writing and intuitive figures.

Weaknesses
  - Clean training data assumption: The method assumes training data contains only normal samples, which may not hold in real-world scenarios. Robustness to label noise or contaminated training sets is not evaluated.
  - Limited handling of categorical/text data: The paper does not discuss whether the Gaussian noise and latent space assumptions remain valid for categorical data. Free-text data handling is not addressed.

---

> ### Author Rebuttal · Authors · 2026-03-31
>
> We thank the reviewer for their valuable feedback and strong support
> of our work.
>
> $\textbf{(Q1) Contaminated Training Data:}$ We agree that real-world
> anomaly detection often involves mildly contaminated training data.
> To evaluate robustness, we conducted additional experiments on 4 datasets
> ($\textit{Annthyroid}$, $\textit{OS}$, $\textit{DAMRE}$, and $\textit{SMD}$)
> with contamination ratios ranging from 0\% to 5\% in the training
> set. As shown in the generated plot (https://shorturl.at/uGIRR),
> CausalAno maintains stable performance across all contamination levels
> and consistently outperforms the strongest baseline DRL. Intuitively,
> the causal factorization introduces a structural inductive bias: the
> generator models conditional relationships through low-dimensional
> sub-generators $G_{i}$, which reduces its tendency to fit arbitrary
> joint patterns induced by a small fraction of anomalous samples. In
> addition, anomaly scoring is performed in the learned latent space,
> where the influence of a small number of contaminated points is limited
> when estimating the mean and covariance. We will add these robustness
> experiments to the revised manuscript.
>
> $\textbf{(Q2) Text Handling and Gaussian Assumptions:}$ We appreciate
> the opportunity to clarify the scope and data-type handling of our
> framework.
>
> $\textit{Free-Text Attributes:}$ Our framework focuses on $\textit{structured
> tabular data}$ (continuous and categorical variables), consistent with
> recent tabular AD methods such as DTE (ICLR 2024), DRL (ICLR
> 2025), and LLM-DAS (ICLR 2026). Handling raw free-text requires
> additional representation learning (e.g., pretrained language model
> embeddings), which is outside the scope of this work. In principle,
> such embeddings could be used as inputs to CausalAno, but we leave
> this extension for future work and will clarify this scope explicitly
> in the revision.
>
> $\textit{Categorical Data and Gaussian Assumption:}$ Categorical variables
> are handled via continuous relaxations (e.g., Gumbel-Softmax), allowing
> both discrete and continuous features to be mapped into a shared latent
> space through the feature extractor $h(x)$. We clarify that the Gaussian
> assumption is not imposed on the input space, but on the learned latent
> representation used for anomaly scoring. This assumption is an approximation
> commonly used in representation-based AD. While it is not guaranteed
> to hold exactly, we empirically observe that the learned embeddings
> of normal samples exhibit sufficient concentration for Mahalanobis-based
> scoring to be effective. This is supported by strong performance on
> datasets with predominantly categorical features (e.g., $\textit{Bank}$
> and $\textit{CMC}$ in $\textbf{Table 2}$).

---

> > ### Author Rebuttal · Reviewer_PgYS · 2026-04-03
> >
> > Thanks for the detailed response that answers my questions.

---

> > > ### Author Response · Authors · 2026-04-03
> > >
> > > We sincerely thank the reviewer for their time, thoughtful comments, and for confirming that all concerns were fully addressed.

---

### Decision · Program_Chairs · 2026-04-30

**Decision:**

Accept (regular)

**Comment:**

Based on the reviews, I recommend acceptance. The reviewers agreed that this is a technically solid and well-motivated paper, and the post-rebuttal discussion converged to a clear positive consensus. The proposed causal-DAG-guided GAN provides a coherent approach to detecting tabular anomalies that violate conditional mechanisms rather than merely appearing as marginal outliers.

The main concerns involved dependence on inferred graph quality, the stability and computational cost of the SHD+REINFORCE regularizer, the gap between the theory and the implemented objective, handling of mixed/categorical features, and robustness to contaminated training data or small datasets. The rebuttal adequately addressed these points, and all three reviewers indicated that their concerns were fully resolved. The final version should incorporate these clarifications and appropriately scope claims that depend on causal discovery quality and the tabular-data setting.